# CONFORMAL PREDICTION ADAPTIVE TO UNKNOWN SUBPOPULATION SHIFTS

## ABSTRACT

Conformal prediction is widely used to equip black-box machine learning models with uncertainty quantification, offering formal coverage guarantees under exchangeable data. However, these guarantees fail when faced with subpopulation shifts, where the test environment contains a different mix of subpopulations than the calibration data. In this work, we focus on *unknown* subpopulation shifts where we are not given group-information i.e. the subpopulation labels of datapoints have to be inferred. We propose new methods that provably adapt conformal prediction to such shifts, ensuring valid coverage without explicit knowledge of subpopulation structure. While existing methods in similar setups assume perfect subpopulation labels, our framework explicitly relaxes this requirement and characterizes conditions where formal coverage guarantees remain feasible. Further, our algorithms scale to high-dimensional settings and remain practical in realistic machine learning tasks. Extensive experiments on vision (with vision transformers) and language (with large language models) benchmarks demonstrate that our methods reliably maintain coverage and effectively control risks in scenarios where standard conformal prediction fails.

## 1 INTRODUCTION

In high-stakes real-world applications of machine learning, such as healthcare, uncertainty quantification (UQ) is crucial to safeguard patient health from the risks posed by model uncertainty. Conformal prediction (CP) techniques (Vovk et al., 2005) offer a framework for uncertainty quantification before model deployment. Formally, conformal prediction guarantees marginal coverage, meaning that for a given input $X_{\text{test}}$ with unknown label $Y_{\text{test}}$ and a user-defined error rate $\alpha$, the probability that $Y_{\text{test}}$ lies in the prediction set $C_\alpha(X_{\text{test}})$ is at least $1 - \alpha$, i.e.,

$$Pr(Y_{\text{test}} \in C_\alpha(X_{\text{test}})) \geq 1 - \alpha, \ \text{for } (X_{\text{test}}, Y_{\text{test}}) \sim \mathbb{P}_{\text{test}} \,. \tag{1}$$

The size of the prediction set $C_\alpha(X_{\text{test}})$ reflects the level of uncertainty–larger sets indicate higher uncertainty, while smaller sets signal greater confidence. The threshold used in conformal prediction determines how conservative the prediction set is, balancing between coverage and uncertainty.

Standard conformal prediction offers provable marginal coverage guarantees under the assumption that test data is exchangeable with the training data. However, in many real-world scenarios, this assumption is violated due to distribution shifts. One of the most common types of distribution shift is subpopulation shift, where the proportions of subpopulations differ between training and deployment environments (Yang et al., 2023). A key challenge arises when different subpopulations present varying levels of prediction difficulty, requiring distinct thresholds to maintain reliable marginal coverage across all subpopulations. Distribution shifts, particularly subpopulation shifts, complicate this task further by causing the proportions of subpopulations to differ between training and test environments. As a result, a uniform threshold might not provide adequate marginal coverage for all subpopulations.

To address this, we propose a two-stage approach. First, we train a classifier that, given a test input $X$, predict a probability distribution over the subpopulations $X$ belong to. We will refer to this classifier as the *domain classifier*. We then use the predicted probabilities to weigh the calibration data to adapt the threshold for conformal prediction accordingly, ensuring the prediction set reflects the uncertainty appropriate for each subpopulation.

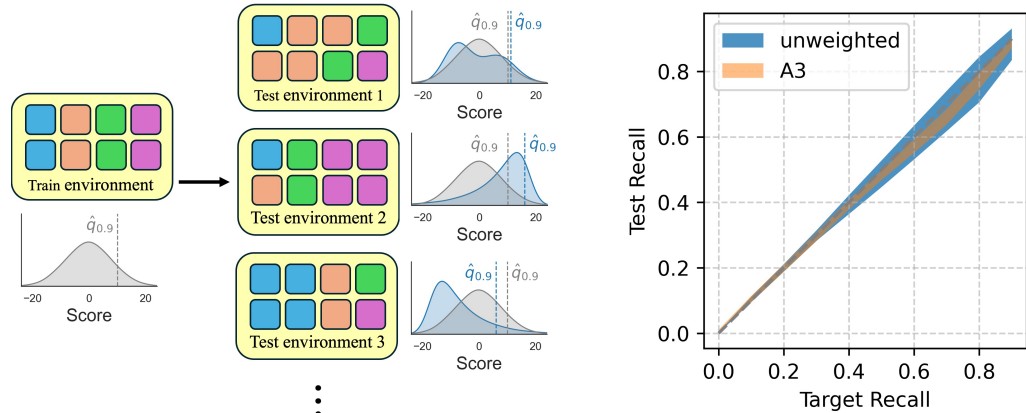

Figure 1: (Left) Example of subpopulation shifts with 4 domains and 3 test environments. Each colored square represents data from a particular domain. Train and test environments are mixtures of the same set of domains but at different proportion. Score distributions (gray for train environment and blue for each test environment) and threshold calculated from standard conformal prediction are shown for each train/test environment. Subpopulation shifts leads to roughly the ideal coverage in test environment 1, whereas shifts for test environment 2 and 3 lead to significant under and over-coverage respectively. (Right) The same issue arises in LLM hallucination detection across different test environments. Standard LLM uncertainty estimation method (blue) is sensitive to distribution shifts displaying high variance in its hallucination detection recall, while the recall with our modification (orange) tightly follows the desired target recall.

**Key contributions.** We make these contributions to the problem of adapting conformal prediction to unknown subpopulation shift settings.

- We introduce an algorithm class that is adaptive to arbitrary mixtures over domains by utilizing a *learned domain classifier*. These can be seen as *test-time adaptation* methods that adaptively adjust the conformal prediction threshold. We prove that under mild assumptions, our new algorithms guarantee tight coverage for arbitrary subpopulation shifts.

- We extend the method to when we do not have access to (even imperfect) domain classifier. In this case, we adaptively filter and reweight the calibration data to adaptively pick a threshold for each test data point.

- We run extensive experiments simulating realistic subpopulation shifts on high-dimensional vision classification datasets. We show that our methods consistently provide tight coverage across test environments, unlike prior approaches that under or over cover.

- We also extend our methods to the conformal risk control where we are tasked with controlling the *hallucination risk* in large language models (LLMs). We show that our methods improve upon the state of the art uncertainty estimation for short-form question answering tasks and provide tighter recall under distribution shifts.

## 2 LIMITATIONS OF PRIOR APPROACHES FOR SUBPOPULATION SHIFTS

### 2.1 PRELIMINARIES

We let $\mathcal{X}$ and $\mathcal{Y}$ denote the input and target space of a multiclass classification task. $\hat{f} : \mathcal{X} \rightarrow \Delta^J$ is the pre-trained classifier for the classification task and the output of $\hat{f}$ is a probability distribution over $J$ possible outcomes, e.g., the softmax output of a neural network. $\hat{f}(X)_i$ represents the $i$-th entry of the output of $\hat{f}$. We let $z : \mathcal{X} \rightarrow \mathbb{R}^d$ denote an embedding function that maps the input into a d-dimensional embedding space, and $h : \mathcal{X} \times \mathcal{X} \rightarrow \mathbb{R}$ stands for some similarity measure between embeddings, which make use of $z$ implicitly.

We denote by $\mathbb{P}_k$ the distribution of the $k$-th domain, where there are $K$ domains in total. The calibration dataset from the $k$-th domain is represented as $\{(X_i^k, Y_i^k)\}_{i=1}^{n_k}$, and each pair $(X_i^k, Y_i^k)$ is assumed to be drawn i.i.d. from $\mathbb{P}_k$. The overall calibration dataset is sampled from the training

environment $\mathbb{P}_{\text{train}}$, which is a mixture of the $K$ domain distributions. The score function is represented by $S : \mathcal{X} \times \mathcal{Y} \to \mathbb{R}$ and will make use of $\hat{f}$. We define $m_k(q)$ as $|\{S(X_i^k, Y_i^k; \hat{f}) \le q\}|$, which is the number of number of calibration data in domain $k$ with score less than or equal to $q$.

Lastly, the test data is denoted by $(X_{\text{test}}, Y_{\text{test}})$, drawn from a test environment $\mathbb{P}_{\text{test}}$, which is an unknown mixture of the $k$ domains $\mathbb{P}_k$. We denote the set of all possible test environments by $\mathcal{D}$. Examples of such test environments are illustrated in Figure 1.

## 2.2 Conformal Prediction under Subpopulation Shifts

The standard conformal prediction procedure relies on the exchangeability assumption between calibration and test data. However, in many real-world scenarios—such as dynamic time series—this assumption often does not hold (Prinster et al., 2024). In this work, we focus on the setting of subpopulation shifts. Specifically, we have $K$ domain-specific distributions, denoted by $\mathbb{P}_k$. The test data is sampled i.i.d. from a test environment $\mathbb{P}_{\text{test}}$ such that

$$\mathbb{P}_{\text{test}} = \sum_{k=1}^{K} \lambda_k \mathbb{P}_k, \tag{2}$$

where $\lambda_k$ is the probability that $\mathbb{P}_{\text{test}}$ is drawn from $\mathbb{P}_k$. Importantly, the weights $\lambda_k$'s are *unknown* and arbitarily different from the mixture weights of the calibration data distribution.

**Failure of Standard Conformal Prediction (CP).** In standard CP, (1) is not guaranteed if the test data is not exchangeable with the calibration data due to subpopulation shifts (Tibshirani et al., 2020). For instance, if the test environment has higher probability to be drawn from a harder domain, i.e., $\lambda_k$ is large for domain $k$ where data typically receive higher scores, then standard conformal prediction would result in under-coverage. Conversely, if $\lambda_k$ is large for domain $k$ which has data with lower scores, it would lead to over-coverage. As illustrated in Figure 1, test environment 2 exhibits under-coverage, while test environment 3 demonstrates over-coverage. See App. B for some background on CP.

**CP under Distribution Shifts.** When the distribution shift is known, Tibshirani et al. (2020) showed that we can recover coverage by reweighting calibration score by the covariate likelihood ratio between training and test distribution. However, this approach relies on either knowledge of the test covariate distribution - which in our case is unknown - or estimating this density ratio using a held-out set sampled from the test distribution, which is prohibitive in high-dimensional modern ML. Alternatively, robust or *max* CP proposes to use a fixed threshold that guarantees coverage for a worst-case distribution shift (Cauchois et al., 2024). In our setup, this corresponds to $\mathbb{P}_{\text{test}}$ being drawn only from the "hardest" domain. Suppose $\hat{q}_\alpha^k$ is the domain-specific threshold for $\mathbb{P}_k$ i.e. $\hat{q}_\alpha^k = \lceil (n_k + 1)(1 - \alpha)/n_k \rceil$-quantile of the $n_k$ calibration data from domain $k$. The *max* method returns prediction set using the score threshold $\hat{q}_\alpha := \max_{k \in [K]} \hat{q}_\alpha^k$. However, as we later show, this can be conservative and have significant over-coverage. We want to be adaptive to the actual difficultly of $\mathbb{P}_k$ instead.

**Group Conditional CP.** Group-conditional CP is a closely related approach that has received significant recent attention (Jung et al., 2022; Gibbs et al., 2024; Kiyani et al., 2024; Bairaktari et al., 2025) ,which states that given the input space $\mathcal{X}$ and a collection of groups $\mathcal{G} \subseteq 2^{\mathcal{X}}$, for all $G \in \mathcal{G}$,

$$Pr(Y_{\text{test}} \in C_\alpha(X_{\text{test}})|X_{\text{test}} \in G) \ge 1 - \alpha. \tag{3}$$

This strengthens the standard marginal CP guarantee and simultaneously provides coverage for a collection of subset of $\mathcal{X}$, and is itself a tractable relaxation of exact conditional coverage which is known to be impossible (Barber et al., 2020). Suppose we can satisfy equation 3 for each of the $k$ domains with $\mathcal{G} = [K]$ i.e. for any $G = \mathbb{P}_k$. Such as $C_\alpha$ would also satisfy coverage for our subpopulation shift setting for any weight vector $\lambda$ in equation 2 and hence solves our problem. However, as we next argue, this approach has serious limitations.

## 2.3 Limitations of Conditional Conformal Prediction

**Conditional CP needs group membership.** Indeed, satisfying *group-conditional coverage* implies coverage in our settings, if we define the collection of groups to be the collection of the $K$ domains. However, at test time, group-conditional CP critically needs knowledge of group membership, which is rarely available in practice. Methods have been proposed to learn group memberships with predefined groups (Gibbs et al. (2024), Jung et al. (2022)) or learn the natural partition of the input space

(Kiyani et al., 2024), but there has not yet been an extensive theoretical or practical investigation of what to do when group membership information is imperfect during test time. Similar to our proposed methods, Gibbs et al. (2024) also employs a two-stage approach where they also train a domain classifier. However, their analysis still assumes perfect group information and leaves open the effect of an imperfect domain classifier.

**Conditional CP coverage degrades with imperfect group membership.**

**Theorem 2.1.** *Suppose we are given a algorithm $C_\alpha$ that given perfect group information (whether $X_{test} \sim \mathbb{P}_k$) obtains perfect domain-conditional coverage as in equation 3. Then, there exist domain distributions $\{\mathbb{P}_k\}_{k \in [K]}$, and a domain classifier $c(X_{test}) \in [K]$ with conditional accuracy $\gamma \in [0, 1]$ for every domain $k$, such that if we use $c(X_{test})$ as our imperfect group-information, then*

$$Pr(Y_{test} \in C_\alpha(X_{test})|X_{test} \sim \mathbb{P}_k) \leq \max(0, \gamma - \alpha).$$

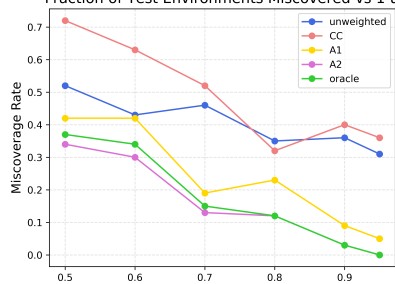

The above theorem shows that the coverage guarantees of any group-conditional conformal predictor can significantly degrade when paired with imperfect group information, demonstrating a big drawback relying in realistic settings where such information is unlikely to be given. This limitation is not just theoretical as we show on the right. We evaluate our methods against the two-stage group-conditional approach (Gibbs et al., 2024) (referred to as *Conditional Calibration (CC)*) on 100 test environments with varying distribution shift and show that it has significantly higher mis-coverage compared with our methods, and is even worse than standard unweighted CP for small $1 - \alpha$.

# 3 SUBPOPULATION SHIFTS WITH AN IMPERFECT DOMAIN CLASSIFIER

## 3.1 WEIGHTED CONFORMAL PREDICTION

To solve the issue caused by distribution shifts, we need to weigh calibration data from each domain differently based on the test environment. For example, if $\lambda_k$ is high, we will need to weigh calibration data from domain $k$ higher since $\mathbb{P}_k$ represents the test environment more closely. We propose training a separate model, $c : \mathcal{X} \to \Delta^K$, named domain classifier, to predict the true $Pr((X_{test}, Y_{test}) \sim \mathbb{P}_k|X_{test})$ for each domain $k$. Then, we will compute the threshold

$$\hat{q} = \text{minimum } q \in \mathbb{R} \text{ such that } \sum_{k=1}^{K} \frac{\hat{\lambda}_k m_k(q)}{n_k + 1} \geq (1 - \alpha), \tag{4}$$

where $\hat{\lambda}_k$ is the $k$-th entry of the output of $c(X_{test})$, $m_k(q)$ is the number of calibration data from domain $k$ with score less than or equal to $q$, and $n_k$ is the number of calibration data from domain $k$. Theorem 3.1 states that if $c$ is a Bayes-optimal classifier, then we get marginal coverage guarantee.

**Theorem 3.1.** *Suppose $c : \mathcal{X} \to \Delta^K$ is a domain classifier that maps the input to a probability distribution over the $K$ domains and $(X_{test}, Y_{test})$ is sampled from $\mathbb{P}_{test}$, as defined in section 2.2. If $c$ is a Bayes-optimal classifier, the output of Algorithm 1, $C_\alpha$, satisfies*

$$Pr(Y_{test} \in C_\alpha(X_{test})) \geq 1 - \alpha.$$

If $\lambda$ is known at test time, we can directly replace $C(X_{test})$ by $\lambda$ and achieve coverage. We refer to this case as the *oracle* method. The proof of Theorem 3.1 can be found in Appendix A.2. Intuitively, if we have a Bayes-optimal domain classifier, the weight given to the domains which are more likely should be higher. In fact, by weighting the calibration scores based on $\lambda$, we can adopt the partial exchangeability proof of Lu et al. (2023) to prove our claim. In the extreme case where the test environment is one of the $K$ in-distribution domains, i.e. $(X_{test}, Y_{test}) \sim \mathbb{P}_{test} = \mathbb{P}_k$, we have that $\hat{\lambda}(X_{test})_i = 1$ for $i = k$ and 0 otherwise. We see that the Algorithm 1 reduces to the case of standard conformal prediction which satisfies (1) since $(X_{test}, Y_{test})$ is now exchangeable with calibration data from domain $k$.

## 3.2 CONFORMAL PREDICTION WITH MULTICALIBRATED DOMAIN CLASSIFIER

In most cases, training a perfect classifier is impossible. Therefore, since $c$ can only provide the estimated probability distribution, how well calibrated $c$ is matters to the coverage provided by

---

**Algorithm 1**

---

**Input** : pre-trained model $\hat{f} : \mathcal{X} \rightarrow \Delta^J$, domain classifier $c : \mathcal{X} \rightarrow \Delta^K$, calibration sets $\{(X_i^k, Y_i^k)\}_{i=1}^{n_k}$ for $k \in [K]$, score function $S : \mathcal{X} \times \mathcal{Y} \rightarrow \mathbb{R}$, error rate $\alpha \in [0,1]$, algorithm type $\in \{''oracle'','' A1''\}$, mixture weight $\lambda \in \Delta^K$ (only if algorithm type is $''oracle''$), test data point $X_{\text{test}}$

**Output** : prediction set $C$

    **for** $k = 1, 2, \cdots, K$ **do**
        **for** $i = 1, 2, \cdots, n_k$ **do**
            $s_i^k \leftarrow S(X_i^k, Y_i^k)$
        **end for**
    **end for**
    **if** algorithm type $== ''oracle''$ **then**
        $\hat{\lambda} \leftarrow \lambda$
    **else if** algorithm type $== ''A1''$ **then**
        $\hat{\lambda} \leftarrow c(X_{\text{test}})$
    **end if**
    Compute $\hat{q}$ following 4
    **return** $\{j \in J | S(X_{test}, j; \hat{f}) \leq \hat{q}\}$             $\triangleright$ $J$ is the number of classes

---

Algorithm 1, especially in cases where the in-domain distributions differ a lot. Therefore, it's more feasible to train a domain classifier that makes mistakes within a limited range. We will use the notion of multicalibration, which is used to measure fairness of a predictor (Hébert-Johnson et al., 2017).

**Definition 3.2.** *[Multicalibrated domain classifier] Denote $\mathcal{D}$ as a family of distributions on $\mathcal{X}$, $c : \mathcal{X} \rightarrow \Delta^K$ as the trained domain classifier, and $c^* : \mathcal{X} \rightarrow \Delta^K$ as the perfect domain classifier. $c$ is multicalibrated with respect to $\mathcal{D}$ if for all $v \in c(\mathcal{X})$ and $D \in \mathcal{D}$,*

$$\mathbb{E}(c^*(x)|x \sim D, c(x) = v) = v.$$

By defining $\mathcal{D}$, as the set of all possible test environments and assuming that the domain classifier, $c$, from Algorithm 1 is multicalibrated with respect to $\mathcal{D}$, we can ensure coverage conditioned on each test environment as shown in Theorem 3.3

**Theorem 3.3.** *[Multicalibrated domain classifier implies coverage under subpopulation shifts] Suppose $c : \mathcal{X} \rightarrow \Delta^K$ is a domain classifier that maps the input to a probability distribution over the $K$ domains and $(X_{test}, Y_{test}) \sim \mathbb{P}_{test}$, as defined in Section 2.2. Furthermore, suppose $\mathcal{D}$ is the set of all possible $\mathbb{P}_{test}$ and $c$ is multicalibrated with respect to $\mathcal{D}$, as defined in Definition 3.2. Then the output of Algorithm 1, $\mathcal{C}_\alpha$, satisfies*

$$Pr(Y_{test} \in C(X_{test})) \geq 1 - \alpha.$$

We refer to Algorithm 1 with a trained domain classifier as the $A1$ method. The proof of Theorem 3.3 can be found in Appendix A.3. While the results of Theorem 3.1 and 3.3 are very similar, they provide coverage guarantee under different assumptions. In Theorem 3.1, we assume that $c$ is a Bayes-optimal classifier which allows us to know $\lambda$ exactly. However, in Theorem 3.3, we made a vastly weaker but sufficient assumption that $c$ is multicalibrated. The assumption allows the true $\lambda$ to be predicted by $c$ on average to recover a similar conditional coverage guarantee.

### 3.3 CONFORMAL PREDICTION WITH MULTIACCURATE DOMAIN CLASSIFIER

While learning multicalibrated predictors is easier than learning the Bayes-optimal classifier, they are still shown to have high computational and sample complexity which makes it difficult to train (Gopalan et al., 2022). Therefore, an even more relaxed assumption is necessary in most cases, which motivates us to use the notion of multiaccuracy (Kim et al., 2018).

**Definition 3.4.** *[Multiaccurate domain classifier] Denote $\mathcal{D}$ as a family of distributions on $\mathcal{X}$, $c : \mathcal{X} \rightarrow \Delta^K$ as the trained domain classifier, and $c^* : \mathcal{X} \rightarrow \Delta^K$ as the perfect domain classifier. $c$ is multiaccurate with respect to $\mathcal{D}$ if for all $D \in \mathcal{D}$,*

$$\mathbb{E}(c^*(x)|x \sim D) = \mathbb{E}(c(x)|x \sim D).$$

---

**Algorithm 2**

---

**Input** : pre-trained model $\hat{f} : \mathcal{X} \rightarrow \Delta^J$, domain classifier $c : \mathcal{X} \rightarrow \Delta^K$, calibration sets $\{(X_i^k, Y_i^k)\}_{i=1}^{n_k}$ for $k \in [K]$, score function $S : \mathcal{X} \times \mathcal{Y} \rightarrow \mathbb{R}$, error rate $\alpha \in [0,1]$, test data set $\{X_{\text{test}}^i\}_{i=1}^{n_{\text{test}}}$

**Outut** :prediction set $C$

    **for** $k = 1, 2, \cdots, K$ **do**
        **for** $i = 1, 2, \cdots, n_k$ **do**
            $s_i^k \leftarrow S(X_i^k, Y_i^k)$
        **end for**
    **end for**
    $\hat{\lambda} \leftarrow \frac{1}{n_{\text{test}}} \sum_{i=1}^{n_{\text{test}}} c(X_{\text{test}}^i)$         ▷ $\hat{\lambda}$ is the average of domain classifier outputs of the test data set

    Compute $\hat{q}$ following 4
    **return** $\{j \in J | S(X_{test}, j; \hat{f}) \leq \hat{q}\}$         ▷ $J$ is the number of classes

---

Under Definition 3.4, multiaccuracy relaxes the definition of multicalibration and only requires a predictor to be calibrated within a subset of $\mathcal{X}$. We propose Algorithm 2 where $\lambda$ is calculated as the average of the outputs of the domain classifier, to adhere to the definition of multiaccuracy which assumes that the output of the trained domain classifier is equal to the ground truth output *in expectation*. By defining the family of subsets, $\mathcal{D}$, as the set of all possible test environments and assuming that $c$ from Algorithm 2 is multiaccurate, we can ensure coverage conditioned on each test environment as shown in Theorem 3.5.

**Theorem 3.5.** *[Multiaccurate domain classifier implies coverage under subpopulation shifts] Suppose $c : \mathcal{X} \rightarrow \Delta^K$ is a domain classifier that maps the input to a probability distribution over the $K$ domains and $(X_{test}, Y_{test}) \sim \mathbb{P}_{test}$, as defined in section 2.2. Furthermore, suppose $\mathcal{D}$ is the set of all possible $\mathbb{P}_{test}$ and $c$ is multiaccurate with respect to $\mathcal{D}$, as defined in Definition 3.4. Then the output of Algorithm 2, $\mathcal{C}_\alpha$, satisfies*

$$Pr(Y_{test} \in C(X_{test})) \geq 1 - \alpha.$$

Comparing to Theorem 3.5 to Theorem 3.3, They provide the same coverage guarantee conditioned on $(X_{\text{test}}, Y_{\text{test}}) \sim \mathbb{P}_{\text{test}}$, however, they differ in assumptions. Theorem 3.5 uses a more relaxed assumption which leads to the change between Algorithm 1 and Algorithm 2. In some sense Algorithm 2 is easier to provide coverage guarantee for because multiaccuracy can be achieved more efficiently.

**Remark 1.** *Multicalibration is difficult to prove formally. However, Hansen et al. (2024) conducted a comprehensive study and show that well trained models tend to be relatively multicalibrated. Thus, we believe that assuming access to a multi-accuracy classifier $c$ (significantly easier to satisfy than multi-calibration) is an easy to satisfy assumption.*

## 4 SUBPOPULATION SHIFTS WITHOUT ANY DOMAIN CLASSIFIER

The two proposed algorithms so far both assume the knowledge of domains at both train and test time, although the exact mixture for the test environments at test time is unknown. To expand on the previous ideas, we empirically study the case where the calibration set, sampled from $\mathbb{P}_{\text{train}}$, is given but we have no knowledge of which of the $K$ domains each calibration data belong to.

### 4.1 CONFORMAL PREDICTION WEIGHTED BY SIMILARITY MEASURES

In many real word tasks, similarity measures in the representation space often capture the semantic similarity between images or languages. Therefore, we propose Algorithm 3 which assumes that data with higher similarities in the embedding space have higher probability to be from the same domain. Algorithm 3 is exactly the weighted conformal prediction method proposed by Tibshirani et al. (2020) where instead of weighing the calibration data by the likelihood ratio, we propose weighing the calibration data by similarity between the embedding of each calibration data and the test data. Weighting by similarity measures assumes that data with high similarity measures are semantically similar, i.e., from the same or similar domains. However, empirical results show that such assumption is not true across all domains, therefore, we propose keeping only a fraction of

---

**Algorithm 3**

---

**Input** : pre-trained model $\hat{f} : \mathcal{X} \to \Delta^J$, calibration sets $\{(X_i, Y_i)\}_{i=1}^n$, score function $S : \mathcal{X} \times \mathcal{Y}$, error rate $\alpha \in [0, 1]$, $\beta \in [0, 1]$, similarity function $h : \mathcal{X} \times \mathcal{X} \to \mathbb{R}$, $\sigma \in \mathbb{R}$, test data $X_{n+1}$
**Output** : prediction set $C$

$n' \leftarrow \lceil \beta n \rceil$
Keep the top $n'$ calibration data, ranked by $h(X_i, X_{n+1})$. The remaining calibration data is denoted by $(X_i', Y_i')$ and the test data is denoted by $(X_{n'+1}', Y_{n'+1}')$
Calculate $s_i \leftarrow S(X_i', Y_i')$ for each calibration data $(X_i', Y_i')$
$s_{n'+1} \leftarrow \infty$
$\gamma_i \leftarrow h(X_{n'+1}', X_i')$ for $i = 1, 2, \cdots, n' + 1$.
$m \leftarrow \text{Softmax}(\{\gamma_i / \sigma\})$
$\hat{q} \leftarrow \text{Quantile}\left(1 - \alpha, \sum_{i=1}^{n'+1} m_i \delta_{s_i}\right)$            ▷ $\delta_{s_i}$ denotes a point mass at $s_i$
$C \leftarrow \{j \in J | S(X_{\text{test}}, j; \hat{f}) \leq \hat{q}\}$
**return** $C$

---

the data with the highest similarity measures to the test data. The percentage of data to include is defined as $\beta$ in Algorithm 3.

### 4.2 Conformal risk control for LLM hallucination detection

The same framework from 4.1 can be extended to make binary decisions, e.g., LLM hallucination detection in short-form question answering tasks. To achieve this, we will use the conformal risk control to lower bound the test recall for detecting hallucination with $r_{\text{test}}$, where hallucinated generations are class 1. Formally, given a test data $(X_{\text{test}}, Y_{\text{test}})$, a target recall $r_{\text{test}}$, we wish to construct $C : \mathcal{Y} \to \{\pm 1\}$ such that

$$\mathbb{E}[Pr(C(Y_{\text{test}}^*) = 1 | A(X_{\text{test}}, Y_{\text{test}}, Y_{\text{test}}^*) = 1)] \geq r_{\text{test}}$$

where $Y_{\text{test}}^*$ is the greedy output to the query $X_{\text{test}}$, $Y_{\text{test}}$ is the ground truth and $A(X_{\text{test}}, Y_{\text{test}}, Y_{\text{test}}^*) = 1$ if $Y_{\text{test}}^*$ is a hallucinated response to query $X_{\text{test}}$ and 0 otherwise. We will follow the steps from Algorithm 3 and make necessary adjustments. Specifically, first, since we wish to bound the recall error, all calibration data are hallucinated generations. Second, we compute the score using score function $S : \mathcal{X} \times \mathcal{Y} \to \mathbb{R}$, which uses a generative model $\hat{f}$. We note that this scoring function is different from the score functions from the vision tasks, as the score does not take the ground truth into account. We then follow the same steps in Algorithm 3 to find the threshold $\hat{q}_\alpha$ where we let $\alpha$ to be $1 - r_{\text{test}}$. For the unweighted baseline, $\hat{q}_\alpha$ is computed as the $\alpha$-quantile of the scores instead. Lastly, we label the test data "hallucination" if the score is above $\hat{q}_\alpha$ and "not hallucination" otherwise.

## 5 Experiments with knowledge of domains

### 5.1 Experimental Setup for vision tasks

**Dataset.** For the vision tasks, we use the ImageNet Large Scale Visual Recognition Challenge dataset (Russakovsky et al., 2015), which contains 1000 classes. We split the validation data in two, half as the calibration set and the other half as the test set. The split is done multiple times as the coverage guarantee of conformal prediction is over the randomness of the calibration set. To simulate subpopulation shifts, we adopt the BREEDS methodology (Santurkar et al., 2020). The method creates a tree structure where the leaf nodes are the 1000 classes and the internal nodes are superclasses. We picked the nodes at level 3 as our domains and the descendents of each node are the classes in each domain. To create a balanced train environment, we keep the number of classes in each domain the same by removing domains with non-sufficient number of classes and removing some classes from domains with too many classes. We test on two different number of classes, one with 26 domains with 3 classes each and the other with 15 domains and 17 classes each. To simulate the different test environments, we follow the sampling strategy from Hsu et al. (2019) to draw $\lambda$ from a Dirichlet distribution with parameter $\alpha'$. The parameter $\alpha'$ controls the heterogeneity, i.e., as $\alpha' \to 0$, $\lambda$ is 1 for one domain and 0 for all others. As $\alpha' \to \infty$, $\lambda$ becomes uniform which reduces the problem to the no subpopulation shift case.

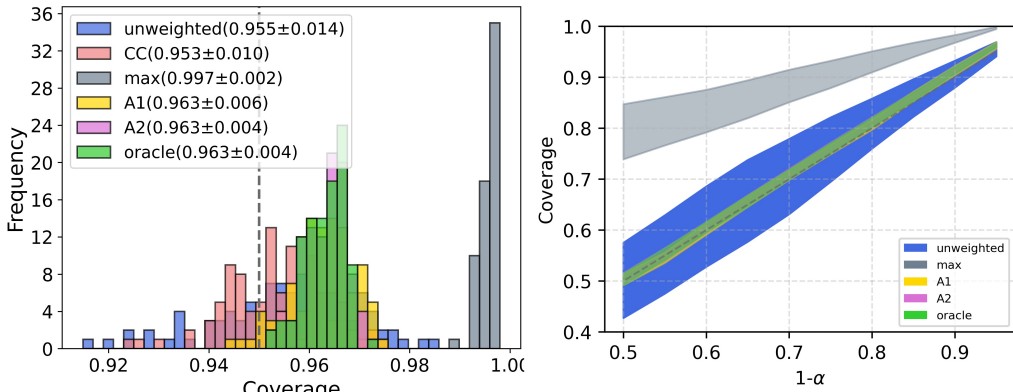

Figure 2: Coverage distribution over 100 test environments with subpopulation shifts. (Left) Coverage across 100 test environments generated using Dirichlet sampling over 26 domains, and the averaged over 15 calibration/test splits. Mean and standard deviations are shown in the legend. (Right) Mean and standard deviation of coverage across 100 test environments. Note that max tends to substantially over cover compared to desired coverage of 0.95. We refer to Algorithm 1 with known $\lambda$ as oracle and Algorithm 1 or 2 with trained domain classifier $c$ as A1 or A2 respectively. Our algorithms (A1, A2, and oracle) demonstrate the desired coverage across test environments (unlike unweighted and Conditional Calibration that have significant under-coverage). They also have minimal over-coverage and tightly follow the target (unlike max which significantly over-covers). Further, the practical algorithms A1 and A2 quite closely match the ideal oracle coverage.

**Models.** We test on three different pretrained models: resnet50 pretrained on ImageNet (He et al., 2015), vision transformer pretrained on ImageNet21k and finetuned on ImageNet 2021 (Steiner et al., 2021; Dosovitskiy et al., 2021; Wightman, 2019), and vision transformer pretrained on WIT-400M image-text pairs by OpenAI using CLIP embedding and finetuned on ImageNet-1k (Radford et al., 2021; Cherti et al., 2022; Dosovitskiy et al., 2021; Wightman, 2019). For the domain classifiers, we modified the fully-connected layers of the three pre-trained models. The modified fully-connected layers now includes three dense layers with sizes 2048, 1024, and 512. The output layer is a softmax layer with output size of either 26 or 15.

**Domain Classifier Training.** For training, only the last 3 fully connected layers are updated. The training uses Adam (Kingma & Ba, 2017) with cross entropy loss. After training, the domain classifiers are then calibrated using Multi-domain temperature scaling introduced in Yu et al. (2022) to reduce calibration error.

## 5.2 MAIN RESULTS

**Coverage with varying test environments.** We calibrated a pre-trained vision transformer with LAC score function and tested it on test set sampled from 100 different test environments. The test environment consists of 26 domains, with 3 classes in each domain while the $\lambda$ was sampled from a Dirichlet distribution with parameter 0.1. Each coverage datapoint is averaged across 15 random calibration/test split. The results are plotted in Figure 2. From Figure 2 (Left) we observe that all three proposed algorithms were able to provide coverage for all test environments while standard conformal prediction could not for some test environments. For the max method, which conformalize the model using the worst case method mentioned in section 2.2, we see that marginal coverage is satisfied for all test environments, however, they are severely over-covered. We also observe that when compared to the standard conformal prediction, the standard deviations for the proposed algorithms are much smaller. This shows the adaptiveness of the proposed algorithms to maintain the desired coverage across test environments. From Figure 2 (Right), we see that the proposed algorithms are able to maintain coverage, while ensuring low standard deviations across different $1 - \alpha$.

**Coverage under different settings.** We obtained the coverage results with varying score functions, model architectures, and degree of subpopulation shifts which we present in Appendix D. The coverage results are consistent across different settings.

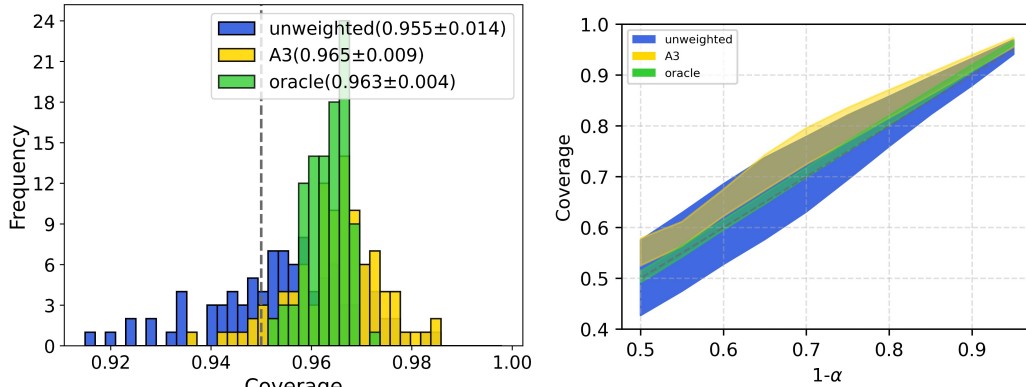

Figure 3: Adapting to subpopulation shifts without a domain classifier. Vision transformer is calibrated with *LAC* score function for various algorithms. For the results of Algorithm 3, the parameters $\sigma$ and $\beta$ are 0.7 and 0.1 respectively. (Left) Coverage across 100 test environments at $\alpha = 0.05$. Each coverage data is the average of 15 calibration/test splits. Mean and standard deviations are shown in the legend. (Right) Mean and standard deviation of coverage across 100 test environments. We refer to Algorithm 1 with known $\lambda$ as oracle and Algorithm 3 as A3. Our algorithm (A3 in pink) demonstrates the desired coverage of 0.95 across test environments with minimal overcoverage. Further, even without using any distributional or domain information, it matches the ideal coverage of the oracle (in green) which knows the test distribution exactly.

## 6 EXPERIMENTS WITHOUT KNOWLEDGE OF DOMAINS

We test our proposed Algorithm 3 with the same settings as Section 5. For the vision tasks, although the same calibration set is used, we do not assume knowledge of the domain label.

### 6.1 EXPERIMENTAL SETUP FOR LANGUAGE TASKS

**Datasets.** To simulate different domains in generative language tasks, we use two distinct datasets: TriviaQA (Joshi et al., 2017), a closed-book question answering dataset, and GSM8K (Cobbe et al., 2021), a mathematical reasoning benchmark. Specifically, we use 2,500 samples from the test split of TriviaQA and the full GSM8K test set, which contains 1,319 questions. To create the calibration and test data, we first randomly select 500 TriviaQA samples and 500 GSM8K samples to create the test set. The rest of the samples are used as the calibration set. To keep the calibration set balanced, we randomly removed 1181 TriviaQA samples, resulting in a calibration set with 1638 samples. We repeat this process 10 times. To simulate each test environment, we again draw $\lambda$ from a dirichlet distribution with parameter 0.1 and remove test data from each of the two domains to match the $\lambda$.

**Models.** We use LLaMA-3-8B (AI@Meta, 2024) as the generative model and obtain responses via greedy decoding. Following prior work (Lin et al., 2024; Bakman et al., 2024), we employ GPT-4o (OpenAI, 2023) as the correctness evaluator, using the query, generated response, and ground truth answer(s) as input. To assess the similarity between test samples and calibration data points, we use the all-mpnet-base-v2 model from SentenceTransformers (Reimers & Gurevych, 2019).

### 6.2 RESULTS FOR VISION TASKS

**Coverage with varying test environments.** We obtain the results for Algorithm 3 with the same setup as section 5.2 and the results are plotted in Figure 3. From Figure 3 (Left) we observe that Algorithm 3 was able to provide coverage for the majority of test environments while standard conformal prediction could not for a significant number of test environments. Although not as small as Algorithm 1 and 2, Algorithm 3 is still able to obtain smaller standard deviation than the standard conformal prediction which shows the adaptiveness of Algorithm 3 even without any knowledge of the domains. From Figure 3 (Right), we see that Algorithm 3 is able to maintain coverage, while ensuring low standard deviations across different $1 - \alpha$.

### 6.3 RESULTS FOR LANGUAGE TASK

We obtain the results for algorithm described in 4.2 and shown in Figure 4. We see that the test recalls follow roughly to the target recall for both standard conformal prediction and the proposed Algorithm 3. However, standard conformal prediction produce results that have larger standard

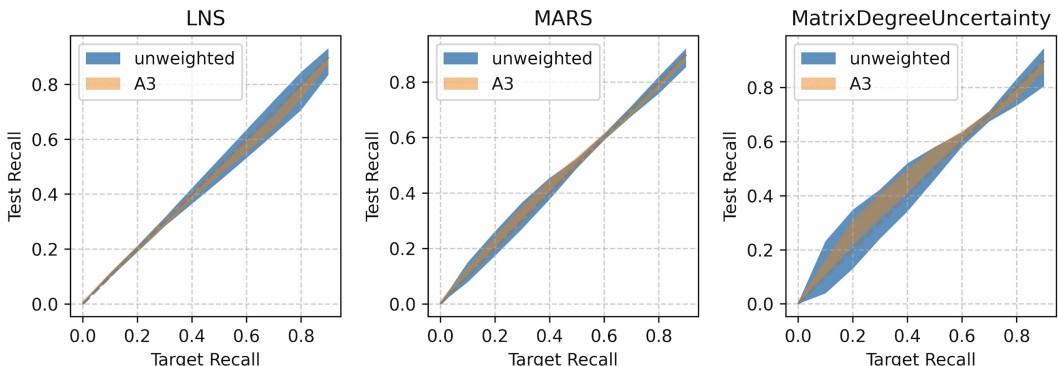

Figure 4: Controlling LLM hallucinations. LlaMA-3-8B was calibrated with 3 different score functions and test data were labeled according to 4.2. Recall was calculated with the standard deviation plotted. The standard deviation is across 100 different test environments, obtained by sampling Dirichlet distribution with $\alpha' = 0.5$. Unweighted LLM uncertainty estimation method (blue) is sensitive to distribution shifts as evidenced by the high variance in recall across test-environments, while the recall with our method A3 (orange) tightly follows the desired target recall.

deviation than Algorithm 3. The results show the necessity of our algorithm for reliable decisions in the hallucination detection task in LLMs under various subpopulation shifts.

## 7 CONCLUSION, LIMITATIONS, AND FUTURE WORK

This paper introduced three algorithms that extended conformal prediction to a setting with subpopulation shifts. For Algorithm 1, we proved that it provides a statistical guarantee to marginal coverage under the assumption that the domain classifier in the algorithm is multicalibrated. Similarly, for Algorithm 2, we proved that it provides marginal coverage under the assumption that the domain classifier is multiaccurate. We evaluated the algorithms experimentally with a synthetic dataset which showed improvement from the standard conformal prediction algorithm in terms of providing coverage when standard conformal prediction did not.

A theoretical limitation of our method is that it does not take advantage of independence between samples from multiple domains which contributes to some over-coverage. This matters when the distribution shift is very mild, as we explore in the Appendix. Improving this is one interesting future work direction. On the practical side, our results do not provide guidance on what score function to pick. Also, our current work explores a single objective - generalizing conformal risk control in LLMs to reliably simultaneously control multiple risks such as hallucination, toxicity, sychophany, etc. is a practically impactful future direction.

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

## A PROOFS

### A.1 PROOF OF THEOREM 2.1

*Proof.* Suppose we have the case of $k = 2$ such that

$$S(X_{\text{test}}) \in \begin{cases} [0, 1) & \text{if } X_{\text{test}} \sim \mathbb{P}_1 \\ [1, 2] & \text{if } X_{\text{test}} \sim \mathbb{P}_2. \end{cases} \tag{5}$$

Furthermore, suppose $q_{\alpha,1}$ and $q_{\alpha,2}$ are the $1-\alpha$ quantile of scores from domain 1 and 2 respectively. In the extreme case that all $1 - \gamma$ fraction of the mistakes that the domain classifier makes are on $X$'s such that $S(X) < q_{\alpha,2}$. It must be that these inputs are mis-covered because $S(X) > q_{\alpha,1}$. Therefore, the fraction of X that are covered is at most $1 - \alpha - (1 - \gamma) = \gamma - \alpha$. $\qquad\square$

### A.2 PROOF OF THEOREM 3.1

This proof follows the proof for Theorem 4.3 from Lu et al. (2023) with some modifications. Suppose $\mathcal{E}$ is the event

$$\mathcal{E} = \{\forall k \in [K], \exists \pi_k, (S^k_{\pi_k(1)}, \cdots S^k_{\pi_k(n_k)}, S^k_{\pi_k(n_k+1)}) = (s^k_1, \cdots, s^k_{n_k}, s^k_{n_k+1})\},$$

where $\{s^k_i\}_{i\in[n_k+1], k\in[K]}$ is the sorted numerical values of the score values. Furthermore, suppose $c$ is a perfect classifier, i.e., $c(X) = c^*(X)$ for all $X \in \mathcal{X}$ where $c^*$ is the true predictor in predicting $\lambda$. Therefore, we have that

$$Pr(S(X_{\text{test}}, Y_{\text{test}}; \hat{f}) \leq \hat{q}_\alpha | \mathcal{E})$$

$$= \sum_{k=1}^{K} \lambda_k Pr(S(X_{\text{test}}, Y_{\text{test}}; \hat{f}) \leq \hat{q}_\alpha | \{S(X^k_1, Y^k_1; \hat{f}), \cdots,$$

$$S(X^k_{n_k}, Y^k_{n_k}; \hat{f}), S(X_{\text{test}}, Y_{\text{test}}; \hat{f})\} \text{ are exchangeable}, \mathcal{E}).$$

Since $S(X^k_1, Y^k_1; \hat{f}), \cdots, S(X^k_{n_k}, Y^k_{n_k}; \hat{f}), S(X_{\text{test}}, Y_{\text{test}}; \hat{f})$ are exchangeable, we have that the above expression is lower bounded by

$$\sum_{k=1}^{K} \frac{\lambda_k m_k(\hat{q}_\alpha)}{n_k + 1},$$

which is lower bounded by $1 - \alpha$ by definition of $\hat{q}_\alpha$. Therefore, we have that

$$Pr(S(X_{\text{test}}, Y_{\text{test}}; \hat{f}) \leq \hat{q}_\alpha | \mathcal{E}) \geq 1 - \alpha.$$

Since this holds for every $(s^k_1, \cdots, s^k_{n_k}, s^k_{n_k+1})$ for all $k \in [K]$, taking the expectation on both sides gives us

$$Pr(S(X_{\text{test}}, Y_{\text{test}}; \hat{f}) \leq \hat{q}_\alpha) \geq 1 - \alpha,$$

which completes the proof.

### A.3 PROOF OF THEOREM 3.3

Suppose $\mathcal{E}$ is the event

$$\mathcal{E} = \{\forall k \in [K], \exists \pi_k, (S^k_{\pi_k(1)}, \cdots S^k_{\pi_k(n_k)}, S^k_{\pi_k(n_k+1)}) = (s^k_1, \cdots, s^k_{n_k}, s^k_{n_k+1})\},$$

where $\{s^k_i\}_{i\in[n_k+1], k\in[K]}$ is the sorted numerical values of the score values. Furthermore, suppose $c$ is multicalibrated with respect to $\mathcal{G}$, the set of all test environments. Therefore, conditioned on $\mathbb{P}_{\text{test}}$, and $c(X_{\text{test}}) = \hat{\lambda}$, we have that $\mathbb{E}(c^*(X_{\text{test}})|c(X_{\text{test}}) = \hat{\lambda}, (X_{\text{test}}, Y_{\text{test}}) \sim \mathbb{P}_{\text{test}}) = \hat{\lambda}$, where $c^*$ is the true predictor in predicting $\lambda$. Combining this property with the partial exchangeable assumption, we have that

$$Pr(S(X_{\text{test}}, Y_{\text{test}}; \hat{f}) \leq \hat{q}_\alpha | (X_{\text{test}}, Y_{\text{test}}) \sim \mathbb{P}_{\text{test}}, c(X_{\text{test}}) = \hat{\lambda}, \mathcal{E})$$

$$= \sum_{k=1}^{K} \hat{\lambda}_k Pr(S(X_{\text{test}}, Y_{\text{test}}; \hat{f}) \leq \hat{q}_\alpha | \{S(X^k_1, Y^k_1; \hat{f}), \cdots,$$

$$S(X^k_{n_k}, Y^k_{n_k}; \hat{f}), S(X_{\text{test}}, Y_{\text{test}}; \hat{f})\} \text{ are exchangeable}, \mathcal{E}).$$

Since $S(X_1^k, Y_1^k; \hat{f}), \cdots, S(X_{n_k}^k, Y_{n_k}^k; \hat{f}), S(X_{\text{test}}, Y_{\text{test}}; \hat{f})$ are exchangeable, we have that the above expression is lower bounded by

$$\sum_{k=1}^{K} \frac{\hat{\lambda}_k m_k(\hat{q}_\alpha)}{n_k + 1},$$

which is lower bounded by $1 - \alpha$ by definition of $\hat{q}_\alpha$. Therefore, we have that

$$Pr(S(X_{\text{test}}, Y_{\text{test}}; \hat{f}) \le \hat{q}_\alpha | (X_{\text{test}}, Y_{\text{test}}) \sim \mathbb{P}_{\text{test}}, c(X_{\text{test}}) = \hat{\lambda}, \mathcal{E}) \ge 1 - \alpha.$$

Since this holds for every $(s_1^k, \cdots, s_{n_k}^k, s_{n_k+1}^k)$ for all $k \in [K]$, taking the expectation on both sides gives us

$$Pr(S(X_{\text{test}}, Y_{\text{test}}; \hat{f}) \le \hat{q}_\alpha | (X_{\text{test}}, Y_{\text{test}}) \sim \mathbb{P}_{\text{test}}, c(X_{\text{test}}) = \hat{\lambda}) \ge 1 - \alpha.$$

Finally, by law of total probability over all possible $c(X_{\text{test}})$ we get that

$$Pr(S(X_{\text{test}}, Y_{\text{test}}; \hat{f}) \le \hat{q}_\alpha | (X_{\text{test}}, Y_{\text{test}}) \sim \mathbb{P}_{\text{test}}) \ge 1 - \alpha,$$

which completes the proof.

### A.4 PROOF OF THEOREM 3.5

Suppose $\mathcal{E}$ is the event

$$\mathcal{E} = \{\forall k \in [K], \exists \pi_k, (S_{\pi_k(1)}^k, \cdots S_{\pi_k(n_k)}^k, S_{\pi_k(n_k+1)}^k) = (s_1^k, \cdots, s_{n_k}^k, s_{n_k+1}^k)\},$$

where $\{s_i^k\}_{i \in [n_k+1], k \in [K]}$ is the sorted numerical values of the score values. Furthermore, suppose $c$ is multiaccurate with respect to $\mathcal{G}$, the set of all test environments. Therefore, conditioned on $\mathbb{P}_{\text{test}}$, we have that $\mathbb{E}(c^*(X_{\text{test}})|(X_{\text{test}}, Y_{\text{test}}) \sim \mathbb{P}_{\text{test}}) = \mathbb{E}(\hat{\lambda}|(X_{\text{test}}, Y_{\text{test}}) \sim \mathbb{P}_{\text{test}})$, where $c^*$ is the true predictor in predicting $\lambda$ and $\hat{\lambda} = c(X_{\text{test}})$. Combining this property with the partial exchangeable assumption, we have that

$$Pr(S(X_{\text{test}}, Y_{\text{test}}; \hat{f}) \le \hat{q}_\alpha | (X_{\text{test}}, Y_{\text{test}}) \sim \mathbb{P}_{\text{test}}, \mathcal{E})$$

$$= \sum_{k=1}^{K} \hat{\lambda}_k Pr(S(X_{\text{test}}, Y_{\text{test}}; \hat{f}) \le \hat{q}_\alpha | \{S(X_1^k, Y_1^k; \hat{f}), \cdots,$$

$$S(X_{n_k}^k, Y_{n_k}^k; \hat{f}), S(X_{\text{test}}, Y_{\text{test}}; \hat{f})\} \text{ are exchangeable}, \mathcal{E}).$$

Since $S(X_1^k, Y_1^k; \hat{f}), \cdots, S(X_{n_k}^k, Y_{n_k}^k; \hat{f}), S(X_{\text{test}}, Y_{\text{test}}; \hat{f})$ are exchangeable, we have that the above expression is lower bounded by

$$\sum_{k=1}^{K} \frac{\hat{\lambda}_k m_k(\hat{q}_\alpha)}{n_k + 1},$$

which is lower bounded by $1 - \alpha$ by definition of $\hat{q}_\alpha$. Therefore, we have that

$$Pr(S(X_{\text{test}}, Y_{\text{test}}; \hat{f}) \le \hat{q}_\alpha | (X_{\text{test}}, Y_{\text{test}}) \sim \mathbb{P}_{\text{test}}, \mathcal{E}) \ge 1 - \alpha.$$

Since this holds for every $(s_1^k, \cdots, s_{n_k}^k, s_{n_k+1}^k)$ for all $k \in [K]$, taking the expectation on both sides gives us

$$Pr(S(X_{\text{test}}, Y_{\text{test}}; \hat{f}) \le \hat{q}_\alpha | (X_{\text{test}}, Y_{\text{test}}) \sim \mathbb{P}_{\text{test}}) \ge 1 - \alpha,$$

which completes the proof.

## B BACKGROUND ON CONFORMAL PREDICTION

**Conformal Prediction Under Exchangeability** Given a test data $X_{\text{test}}$ with unknown label $Y_{\text{test}}$, a calibration set $\{(X_i, Y_i)\}_{i=1}^{n}$, which is distinct from train and test set, and a user defined error rate $\alpha$, the goal of conformal prediction is to build a prediction set $C_\alpha$ that satisfies 1. To conformalize a model to output a valid prediction set, the following procedure is followed: First, a score function $S$ is defined. Second, the threshold $\hat{q}_\alpha$ is computed as the $\frac{\lceil (n+1)(1-\alpha) \rceil}{n}$ quantile of $\{S(X_i, Y_i; \hat{f})\}_{i=1}^{n}$. Lastly, the prediction set $C_\alpha(X_{test})$ is returned such that $C_\alpha(X_{test}) = \{y | S(X_{test}, y; \hat{f}) \le \hat{q}_\alpha\}$. If the calibration data and the test data are drawn i.i.d from the some domain, then $C_\alpha(X_{test})$ satisfies the marginal coverage guarantee due to exchangeability between calibration and test data (Angelopoulos & Bates, 2021). We refer to this method as the standard or unweighted conformal prediction throughout the paper.

**Conformal Risk Control**   The conformal prediction framework can be extended to provide guarantee beyond coverage. Given a prediction set $C(X_{\text{test}})$, a loss function $\ell$ that decreases as $|C(X_{\text{test}})|$ increases, and a user defined error rate $\alpha$, the conformal risk control guarantee is defined as,

$$\mathbb{E}[\ell(C(X_{\text{test}}), Y_{\text{test}})] \leq \alpha \tag{6}$$

(Angelopoulos et al., 2023). Note that the marginal coverage guarantee can be reduced to 6 if we define $\ell$ as the miscoverage loss, i.e., $\ell(C(X_{\text{test}}), Y_{\text{test}}) = \mathbf{1}\{Y_{\text{test}} \notin C(X_{\text{test}})\}$. We refer to Angelopoulos et al. (2023) for the details of conformal risk control. One application for the conformal risk control framework is in large language model (LLM) uncertainty estimation, in particular, hallucination detection. Hallucination refers to when an LLM generate responses that are factually false or inconsistent with the training data. Conformal risk control can be used to select a threshold to determine whether an LLM output is a hallucination or not while maintaining a theoretical bound to metrics such as sensitivity or precision.

## C   OVERVIEW OF SCORE FUNCTIONS

A (conformal) score function maps an input pair $(X, Y)$ to a real-valued score. A larger score indicates less conformity between $(X, Y)$ and other training data. Although conformal prediction algorithms provide marginal coverage guarantees for arbitrary score functions, a poorly designed score function can lead to uninformative prediction sets. For our experiments we explore 3 different score functions for both vision and language tasks.

### C.1   VISION TASKS

We explore the following commonly used score functions for the vision tasks:

- Least Ambiguous Set-valued Classifier (*LAC*) (Sadinle et al., 2018). Given data $(X, y)$ where $y$ is the true label of $X$, define $S(X, y; \hat{f})$ as

$$S(X, y; \hat{f}) = 1 - f(X)_y.$$

- Adaptive Prediction Set (*APS*) (Romano et al., 2020). Given data $(X, y)$ where $y$ is the true label of $X$, define $S(X, y; \hat{f})$ as

$$S(X, y; \hat{f}) = \sum_{i=1}^{k} f(X)_{\pi(i)},$$

where $\pi$ sorts the labels in descending order of label probability given by $f(X)$ and $k = \pi(y)$. In other words, we add up the label probabilities in descending order until we added the true label probability.

- Regularized Adaptive Prediction Set (*RAPS*) (Angelopoulos et al., 2022). Given data $(X, y)$ where $y$ is the true label of $X$, define $S(X, y; \hat{f})$ as

$$S(X, y; \hat{f}) = \left( \sum_{i=1}^{k} f(X)_{\pi(i)} \right) + a * \max(k - b, 0),$$

where $\pi$ sorts the labels in descending order of label probability given by $f(X)$, $k = \pi(y)$, and $(a, b)$ are regularization parameters.

### C.2   LANGUAGE TASKS

We explore the following commonly used score functions for the language tasks:

- Length Normalized Scoring (*LNS*) (Malinin & Gales, 2021). Given a query $X$ and the generated response $\mathbf{y} = \{y_1, y_2, \cdots, y_L\}$ of length $L$, define $S(X, \mathbf{y}; \hat{f})$ as the average log probability of the generated sequence, i.e.,

$$S(X, \mathbf{y}) = \frac{1}{L} \sum_{i=1}^{L} \log Pr[y_i | y_{<i}, X; \hat{f}],$$

where $y_i$ represents the $i$-th token in the sequence and $y_{<i}$ represents the tokens generated before $y_i$.

- Meaning-Aware Response Scoring (*MARS*) (Bakman et al., 2024). Given a query $X$ and the generated response $\mathbf{y} = \{y_1, y_2, \cdots, y_L\}$ of length $L$, define $S(X, \mathbf{y}; \hat{f})$ as

$$S(X, \mathbf{y}; \hat{f}) = \prod_{i=1}^{L} Pr[y_i | y_{<i}, X; \hat{f}]^{w(\mathbf{y}, X, L, i)},$$

where $w$ represents the token weight that emphasize tokens that contribute to answering the query.

- Degree Matrix Uncertainty (Lin et al., 2024). We adopt the uncertainty estimate definition of $Lin\,et\,al.$ (2024) where the score only depends on the query $X$. Given a query $X$ and $m$ generated responses $\mathbf{y}_1, \mathbf{y}_2, \cdots, \mathbf{y}_m$, first, define $W$ as a matrix of pairwise entailment dependencies where $W_{ij}$ represents the entailment dependency between output response $\mathbf{y}_i$ and $\mathbf{y}_j$. Entailment dependencies are calculated by using a Natural Language Inference classifier (He et al., 2021) that classifies generated responses into three classes: entailment, neutral, or contradiction. We then define the degree matrix $D$ as

$$D_{ii} = \sum_{j=1}^{m} W_{ij}.$$

Lastly, the score is defined as

$$\frac{\text{trace}(mI - D)}{m^2}.$$

## D  ADDITIONAL EXPERIMENTS ON ADAPTING TO DISTRIBUTION SHIFTS WITH DOMAIN KNOWLEDGE

Table 1: Coverage at $\alpha = 0.1$ with 26 domains and 3 classes per domain. The results vary over 3 architectures (VisionTransfomer, Resnet50, and Clip) and 3 score functions (*LAC*, *APS*, an *RAPS*). The mean and standard deviation across 100 test environments, sampled from Dirichlet distirbution with $\alpha' = 0.1$, are recorded. For each of the 100 test environments, coverage result is averaged over 15 random calibration/test splits. The results show that the proposed algorithms consistently outperform standard conformal prediction by having lower standard deviations across the 100 test environments.

|  |  | unweighted | oracle | A1 | A2 |
|---|---|---|---|---|---|
| ViT | LAC | $0.905 \pm 0.026$ | $0.912 \pm 0.006$ | $0.910 \pm 0.009$ | $0.912 \pm 0.006$ |
|  | APS | $0.904 \pm 0.021$ | $0.912 \pm 0.005$ | $0.909 \pm 0.006$ | $0.912 \pm 0.005$ |
|  | RAPS | $0.903 \pm 0.016$ | $0.910 \pm 0.008$ | $0.909 \pm 0.008$ | $0.911 \pm 0.007$ |
| Resnet50 | LAC | $0.907 \pm 0.027$ | $0.911 \pm 0.008$ | $0.909 \pm 0.009$ | $0.912 \pm 0.008$ |
|  | APS | $0.905 \pm 0.022$ | $0.910 \pm 0.007$ | $0.907 \pm 0.007$ | $0.911 \pm 0.007$ |
|  | RAPS | $0.903 \pm 0.015$ | $0.908 \pm 0.008$ | $0.904 \pm 0.009$ | $0.909 \pm 0.007$ |
| Clip | LAC | $0.909 \pm 0.023$ | $0.912 \pm 0.007$ | $0.910 \pm 0.007$ | $0.912 \pm 0.007$ |
|  | APS | $0.908 \pm 0.021$ | $0.913 \pm 0.008$ | $0.910 \pm 0.007$ | $0.914 \pm 0.008$ |
|  | RAPS | $0.902 \pm 0.014$ | $0.910 \pm 0.005$ | $0.909 \pm 0.006$ | $0.910 \pm 0.005$ |

Table 2: Prediction set size for experiment from Table 1

|  |  | unweighted | oracle | A1 | A2 | max |
|---|---|---|---|---|---|---|
| ViT | LAC | 1.311 | 1.396 | 1.501 | 1.3946 | 3.287 |
|  | APS | 107.688 | 112.125 | 110.938 | 112.000 | 184.000 |
|  | RAPS | 3.055 | 3.367 | 3.436 | 3.359 | 8.172 |
| Resnet50 | LAC | 1.793 | 2.006 | 2.297 | 2.004 | 6.121 |
|  | APS | 162.000 | 178.250 | 182.250 | 178.500 | 485.250 |
|  | RAPS | 8.008 | 8.750 | 8.992 | 8.750 | 21.938 |
| Clip | LAC | 1.435 | 1.542 | 1.731 | 1.542 | 4.141 |
|  | APS | 183.625 | 188.875 | 182.500 | 189.000 | 356.000 |
|  | RAPS | 2.961 | 3.285 | 3.516 | 3.287 | 3.285 |

Table 3: Coverage at $\alpha = 0.1$ with 26 domains and 3 classes per domain. The results vary over 3 architectures (VisionTransfomer, Resnet50, and Clip) and 3 score functions (*LAC*, *APS*, an *RAPS*). The mean and standard deviation across 100 test environments, sampled from Dirichlet distirbution with $\alpha' = 1$, are recorded. For each of the 100 test environments, coverage result is averaged over 15 random calibration/test splits. The results show that the proposed algorithms consistently outperform standard conformal prediction by having lower standard deviations across the 100 test environments. The results also show that the difference between standard deviations of standard and the proposed methods are much smaller than those from Table 1. This is a limitation to our proposed algorithms which do not assume independence between data from different domains, leading to more conservative bounds for coverage in this case where the subpopulation shifts are milder (larger $\alpha'$).

|  |  | unweighted | oracle | A1 | A2 |
|---|---|---|---|---|---|
| ViT | LAC | $0.899 \pm 0.011$ | $0.912 \pm 0.003$ | $0.910 \pm 0.004$ | $0.912 \pm 0.003$ |
|  | APS | $0.900 \pm 0.007$ | $0.912 \pm 0.003$ | $0.908 \pm 0.003$ | $0.912 \pm 0.003$ |
|  | RAPS | $0.900 \pm 0.007$ | $0.912 \pm 0.004$ | $0.908 \pm 0.003$ | $0.912 \pm 0.003$ |
| Resnet50 | LAC | $0.899 \pm 0.011$ | $0.912 \pm 0.003$ | $0.907 \pm 0.004$ | $0.913 \pm 0.003$ |
|  | APS | $0.902 \pm 0.008$ | $0.913 \pm 0.003$ | $0.908 \pm 0.004$ | $0.914 \pm 0.003$ |
|  | RAPS | $0.898 \pm 0.006$ | $0.911 \pm 0.004$ | $0.905 \pm 0.003$ | $0.911 \pm 0.003$ |
| Clip | LAC | $0.901 \pm 0.010$ | $0.913 \pm 0.003$ | $0.910 \pm 0.003$ | $0.913 \pm 0.003$ |
|  | APS | $0.904 \pm 0.008$ | $0.916 \pm 0.004$ | $0.910 \pm 0.003$ | $0.916 \pm 0.003$ |
|  | RAPS | $0.900 \pm 0.005$ | $0.911 \pm 0.003$ | $0.908 \pm 0.003$ | $0.911 \pm 0.003$ |

Table 4: Prediction set size for experiment from Table 3

|  |  | unweighted | oracle | A1 | A2 | max |
|---|---|---|---|---|---|---|
| ViT | LAC | 1.318 | 1.406 | 1.543 | 1.405 | 3.389 |
|  | APS | 106.375 | 113.750 | 111.063 | 113.625 | 181.375 |
|  | RAPS | 3.111 | 3.420 | 3.529 | 3.414 | 8.180 |
| Resnet50 | LAC | 1.814 | 2.016 | 2.424 | 2.023 | 6.336 |
|  | APS | 159.000 | 178.875 | 184.875 | 179.375 | 480.000 |
|  | RAPS | 7.996 | 8.852 | 9.188 | 8.875 | 21.781 |
| Clip | LAC | 1.446 | 1.573 | 1.812 | 1.574 | 4.277 |
|  | APS | 179.750 | 193.625 | 185.250 | 193.25 | 350.500 |
|  | RAPS | 3.004 | 3.285 | 3.600 | 3.289 | 8.618 |

Table 5: Coverage at $\alpha = 0.1$ with 15 domains and 17 classes per domain. The results vary over 3 architectures (VisionTransfomer, Resnet50, and Clip) and 3 score functions (*LAC*, *APS*, an *RAPS*). The mean and standard deviation across 100 test environments, sampled from Dirichlet distirbution with $\alpha' = 0.1$, are recorded. For each of the 100 test environments, coverage result is averaged over 15 random calibration/test splits. The results show that the proposed algorithms consistently outperform standard conformal prediction by having lower standard deviations across the 100 test environments. Compared to the results from Table 1, the standard deviations are lower across all algorithms and the mean is much closer to the desired 0.9. The larger number of calibration data here results in a tighter coverage distribution due to the randomness of marginal coverage guarantee for conformal prediction algorithms.

| | | unweighted | oracle | A1 | A2 |
|---|---|---|---|---|---|
| ViT | *LAC* | $0.902 \pm 0.024$ | $0.901 \pm 0.003$ | $0.900 \pm 0.005$ | $0.901 \pm 0.003$ |
| | *APS* | $0.902 \pm 0.015$ | $0.905 \pm 0.003$ | $0.904 \pm 0.003$ | $0.904 \pm 0.003$ |
| | *RAPS* | $0.901 \pm 0.009$ | $0.902 \pm 0.003$ | $0.902 \pm 0.003$ | $0.902 \pm 0.003$ |
| Resnet50 | *LAC* | $0.901 \pm 0.028$ | $0.902 \pm 0.004$ | $0.901 \pm 0.006$ | $0.901 \pm 0.005$ |
| | *APS* | $0.900 \pm 0.026$ | $0.902 \pm 0.004$ | $0.901 \pm 0.004$ | $0.902 \pm 0.004$ |
| | *RAPS* | $0.900 \pm 0.019$ | $0.902 \pm 0.003$ | $0.901 \pm 0.004$ | $0.901 \pm 0.004$ |
| Clip | *LAC* | $0.902 \pm 0.023$ | $0.901 \pm 0.004$ | $0.901 \pm 0.005$ | $0.901 \pm 0.004$ |
| | *APS* | $0.900 \pm 0.024$ | $0.902 \pm 0.003$ | $0.901 \pm 0.003$ | $0.902 \pm 0.003$ |
| | *RAPS* | $0.901 \pm 0.011$ | $0.900 \pm 0.003$ | $0.900 \pm 0.003$ | $0.900 \pm 0.003$ |

Table 6: Prediction set size for experiment from Table 5

| | | unweighted | oracle | A1 | A2 | max |
|---|---|---|---|---|---|---|
| ViT | *LAC* | 1.141 | 1.164 | 1.206 | 1.162 | 1.480 |
| | *APS* | 109.375 | 111.438 | 111.063 | 113.250 | 145.0 |
| | *RAPS* | 2.762 | 2.789 | 2.818 | 2.787 | 3.576 |
| Resnet50 | *LAC* | 1.443 | 1.544 | 1.639 | 1.529 | 2.424 |
| | *APS* | 170.625 | 173.625 | 169.875 | 173.125 | 276.250 |
| | *RAPS* | 7.867 | 7.977 | 7.988 | 7.949 | 12.453 |
| Clip | *LAC* | 1.205 | 1.219 | 1.254 | 1.218 | 1.523 |
| | *APS* | 180.250 | 182.375 | 181.375 | 181.750 | 284.750 |
| | *RAPS* | 2.535 | 2.582 | 2.635 | 2.580 | 3.512 |

Table 7: Coverage at $\alpha = 0.1$ with 15 domains and 7 classes per domain. The results vary over 3 architectures (VisionTransfomer, Resnet50, and Clip) and 3 score functions (*LAC*, *APS*, an *RAPS*). The mean and standard deviation across 100 test environments, sampled from Dirichlet distirbution with $\alpha' = 1$, are recorded. For each of the 100 test environments, coverage result is averaged over 15 random calibration/test splits. The results show that the proposed algorithms consistently outperform standard conformal prediction by having lower standard deviations across the 100 test environments. The results also show that the difference between standard deviations of standard and the proposed methods are much smaller than those from Table 5 due to the limitations of the proposed algorithms.

| | | unweighted | oracle | A1 | A2 |
|---|---|---|---|---|---|
| ViT | *LAC* | $0.900 \pm 0.009$ | $0.901 \pm 0.002$ | $0.899 \pm 0.002$ | $0.901 \pm 0.002$ |
| | *APS* | $0.905 \pm 0.006$ | $0.905 \pm 0.001$ | $0.905 \pm 0.001$ | $0.905 \pm 0.001$ |
| | *RAPS* | $0.901 \pm 0.003$ | $0.903 \pm 0.002$ | $0.902 \pm 0.002$ | $0.903 \pm 0.002$ |
| Resnet50 | *LAC* | $0.902 \pm 0.010$ | $0.902 \pm 0.002$ | $0.901 \pm 0.002$ | $0.902 \pm 0.002$ |
| | *APS* | $0.903 \pm 0.010$ | $0.903 \pm 0.002$ | $0.901 \pm 0.002$ | $0.903 \pm 0.002$ |
| | *RAPS* | $0.901 \pm 0.007$ | $0.902 \pm 0.002$ | $0.901 \pm 0.002$ | $0.902 \pm 0.002$ |
| Clip | *LAC* | $0.899 \pm 0.009$ | $0.901 \pm 0.002$ | $0.901 \pm 0.002$ | $0.901 \pm 0.002$ |
| | *APS* | $0.902 \pm 0.008$ | $0.902 \pm 0.001$ | $0.901 \pm 0.001$ | $0.902 \pm 0.001$ |
| | *RAPS* | $0.900 \pm 0.004$ | $0.902 \pm 0.002$ | $0.900 \pm 0.002$ | $0.901 \pm 0.002$ |

Table 8: Prediction set size for experiment from Table 7

| | | unweighted | oracle | A1 | A2 | max |
|---|---|---|---|---|---|---|
| ViT | *LAC* | 1.146 | 1.165 | 1.224 | 1.163 | 1.494 |
| | *APS* | 109.375 | 110.500 | 109.688 | 110.438 | 145.000 |
| | *RAPS* | 2.779 | 2.824 | 2.857 | 2.822 | 3.598 |
| Resnet50 | *LAC* | 1.455 | 1.502 | 1.656 | 1.493 | 2.451 |
| | *APS* | 169.375 | 175.000 | 169.625 | 174.500 | 275.000 |
| | *RAPS* | 7.793 | 7.908 | 7.984 | 7.953 | 12.352 |
| Clip | *LAC* | 1.214 | 1.229 | 1.275 | 1.229 | 1.540 |
| | *APS* | 179.000 | 181.875 | 180.000 | 181.500 | 283.000 |
| | *RAPS* | 2.555 | 2.598 | 2.676 | 2.596 | 3.531 |

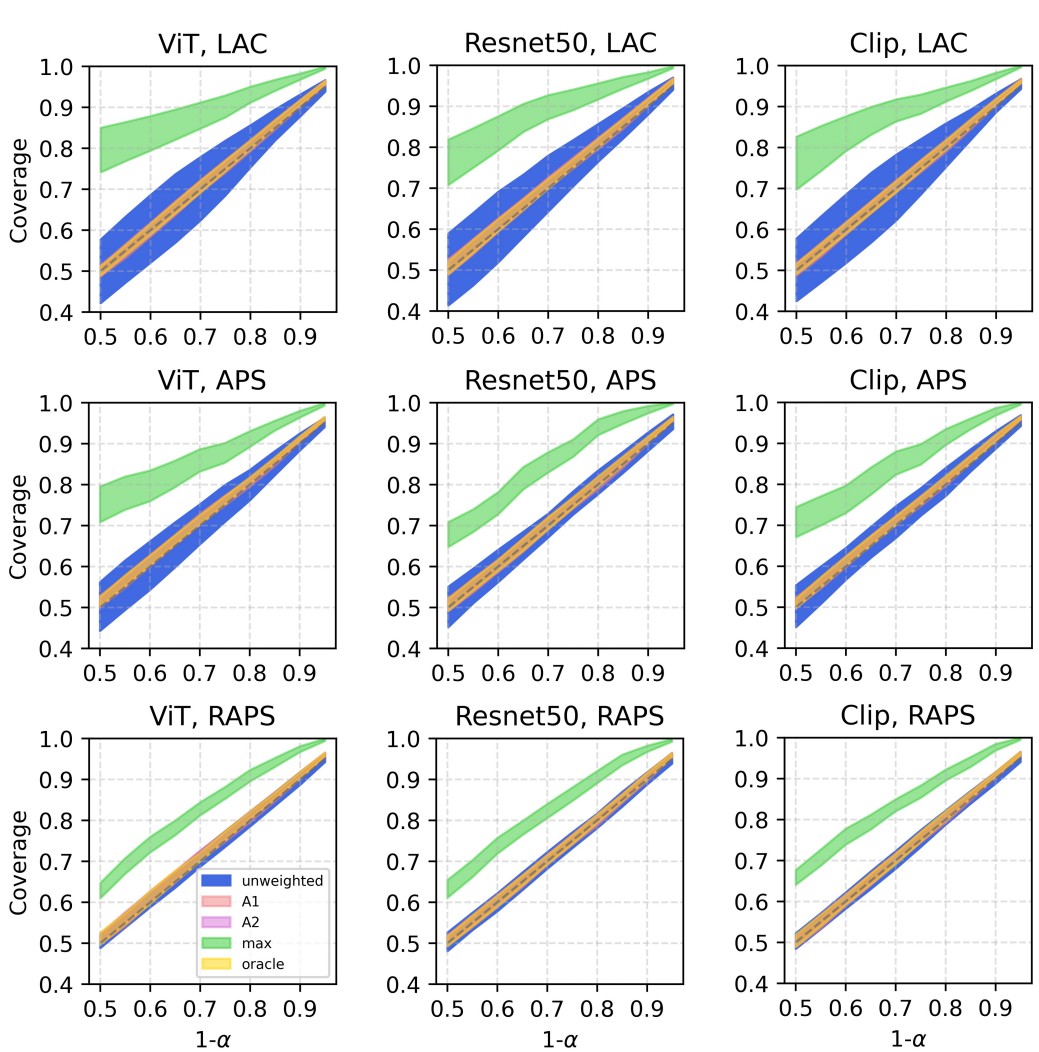

Figure 5: Distribution of coverage across different $1 - \alpha$. The results from 3 different model architectures (VisionTransformer, Resnet50, and Clip) and 3 different score functions (*LAC*, *APS*, and *RAPS*) are shown. For each sub-figure, the standard deviation across 100 test environments, sampled from Dirichlet distribution with $\alpha' = 0.1$, is plotted. For each test environment, the coverage result is the average of 15 random calibration/test splits. The domain structure consists of 26 domains and 3 classes per domain. The results show that the proposed algorithms consistently outperform standard conformal prediction by having lower standard deviations across all model architectures, score functions, and $\alpha$.

# E ADDITIONAL EXPERIMENTS ON ADAPTING TO DISTRIBUTION SHIFTS WITHOUT DOMAIN KNOWLEDGE

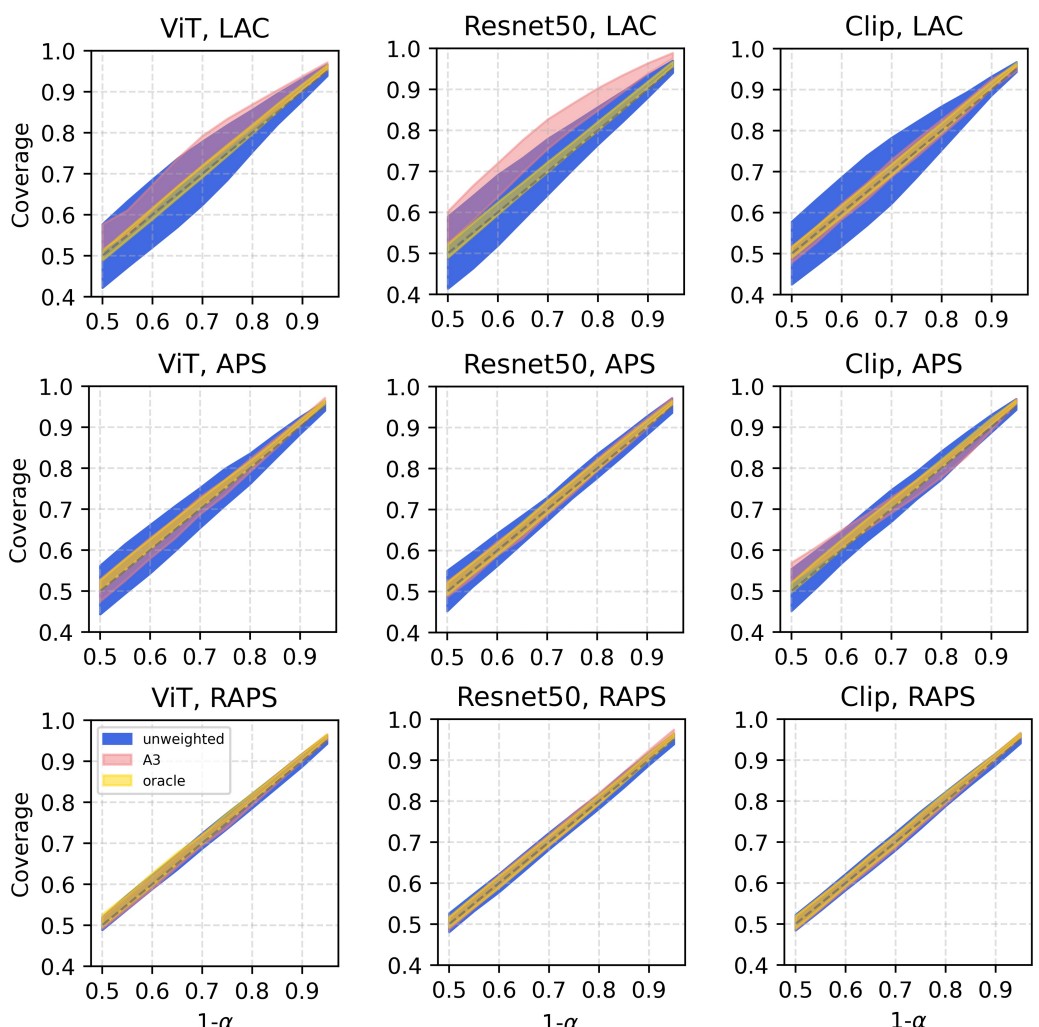

Figure 6: Distribution of coverage across different $1 - \alpha$ for Algorithm 3. For all settings, the top 5% of calibration data were selected and the temperature parameter $\sigma$ was optimized for each $\alpha$ value as shown in Table 9. The results from 3 different model architectures (VisionTransformer, Resnet50, and Clip) and 3 different score functions (*LAC*, *APS*, and *RAPS*) are shown. For each sub-figure, the standard deviation across 100 test environments, sampled from Dirichlet distribution with $\alpha' = 0.1$, is plotted. For each test environment, the coverage result is the average of 15 random calibration/test splits. The domain structure consists of 26 domains and 3 classes per domain. The results show that the proposed algorithms consistently outperform standard conformal prediction by having lower standard deviations across all model architectures, score functions, and $\alpha$. Smaller $\sigma$ values show a lower standard deviation across the 100 test environments, however, it deviates the mean from the ideal $1 - \alpha$ coverage slightly. Conversely, larger $\sigma$ results in larger standard deviation since Algorithm 3 reduces to the unweighted case as $\sigma \to \infty$. Therefore, choosing $\sigma$ is a trade-off between mean and standard deviation across test environments.

Table 9: Prameter ($\sigma$) used to generate results from Figure 6

| | $\alpha$ | 0.05 | 0.1 | 0.15 | 0.2 | 0.25 | 0.3 | 0.35 | 0.4 | 0.45 | 0.5 |
|---|---|---|---|---|---|---|---|---|---|---|---|
| | *LAC* | 2.05 | 1.65 | 1.30 | 1.00 | 0.75 | 0.55 | 0.40 | 0.30 | 0.25 | 0.20 |
| ViT | *APS* | 0.70 | 0.70 | 0.70 | 0.70 | 0.70 | 0.55 | 0.55 | 0.50 | 0.50 | 0.45 |
| | *RAPS* | 1.00 | 0.70 | 0.70 | 0.70 | 0.70 | 0.50 | 0.50 | 0.50 | 0.45 | 0.45 |
| | *LAC* | 0.70 | 0.70 | 0.70 | 0.70 | 0.70 | 0.70 | 0.70 | 0.70 | 0.70 | 0.70 |
| Resnet50 | *APS* | 1.50 | 1.50 | 1.00 | 0.70 | 0.60 | 0.50 | 0.50 | 0.40 | 0.40 | 0.35 |
| | *RAPS* | 2.00 | 2.00 | 1.50 | 1.50 | 1.00 | 0.80 | 0.70 | 0.70 | 0.70 | 0.70 |
| | *LAC* | 2.00 | 0.55 | 0.45 | 0.36 | 0.30 | 0.26 | 0.24 | 0.22 | 0.21 | 0.20 |
| Clip | *APS* | 0.70 | 0.70 | 0.70 | 0.70 | 0.70 | 0.70 | 0.70 | 0.70 | 0.70 | 0.70 |
| | *RAPS* | 0.70 | 0.70 | 0.70 | 0.70 | 0.70 | 0.70 | 0.70 | 0.70 | 0.70 | 0.70 |

## F  INFLUENCE OF DOMAIN CLASSIFIER CALIBRATION ERROR

In this section, we analyze the feasibility of the multicalibration/multiaccuracy assumption of the methods in real-world scenarios. The mean and max ECE for each classifier across 100 different test environments are computed and presented in Table 10. We see that the three domain classifier architecture that we have used in the previous sections exhibit different expected calibration error (ECE). To see the effect of calibration error on the coverage results, we conducted an experiment where the domain classifier $c$ and predictor $\hat{f}$ have different architectures. The results with Vision Transformer as the pretrained predictor architecture are presented in Table 11. Despite the large different in expected calibration error ( 0.01 for mean ECE and 0.02 for max ECE), we see negligible difference in coverage for both Algorithm 1 and Algorithm 2.

Table 10: Domain Classifier ECE

| Architecture | mean ECE | max ECE |
|---|---|---|
| VIT | 0.0326 | 0.0962 |
| Resnet50 | 0.0401 | 0.0745 |
| Clip | 0.0317 | 0.0752 |

Table 11: Coverage at $\alpha = 0.1$ with 26 domains and 3 classes per domain with different domain classifier architectures while fixing $f$ as the Vision Transformer pretrained model

| Domain Classifier Architecture | Score Function | A1 | A2 |
|---|---|---|---|
| | *LAC* | $0.913 \pm 0.009$ | $0.914 \pm 0.007$ |
| ViT | *APS* | $0.911 \pm 0.007$ | $0.913 \pm 0.006$ |
| | *RAPS* | $0.905 \pm 0.007$ | $0.909 \pm 0.007$ |
| | *LAC* | $0.912 \pm 0.009$ | $0.915 \pm 0.008$ |
| Resnet50 | *APS* | $0.911 \pm 0.007$ | $0.913 \pm 0.007$ |
| | *RAPS* | $0.905 \pm 0.007$ | $0.910 \pm 0.006$ |
| | *LAC* | $0.913 \pm 0.008$ | $0.914 \pm 0.007$ |
| Clip | *APS* | $0.911 \pm 0.007$ | $0.913 \pm 0.006$ |
| | *RAPS* | $0.905 \pm 0.006$ | $0.910 \pm 0.007$ |

## G  ADDITIONAL DISCUSSION FOR ALGORITHM 3

In this section, we will discuss the motivation for the design of Algorithm 3.

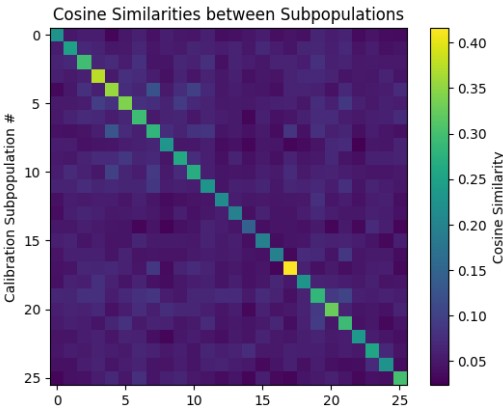

Figure 7: Average cosine similarities between embeddings of calibration data and test data. We observe that the calibration data embeddings from a domain has higher average similarities with test data embeddings from that same domain.

### G.1 EMBEDDING SIMILARITIES

In many real world scenarios, similarity measures in the embedding space often measure semantic similarities between images or languages, for example, recommender systems or word2vec. Indeed, for ImageNet, we observe that images from the same domain have higher cosine similarities with each other. The evidence is shown in Figure 7.

### G.2 HYPERPARAMETERS

For Algorithm 3, we introduced two hyperparameters: $\beta$ and $\sigma$. $\beta$ signifies how many calibration data to select for the CP task. From Figure 8, we see that for test data from subpopulation 1, similarities to calibration data from subpopulation 1 are mostly distributed in the top 5% of all calibration data. Therefore, we introduced $\beta$ in hope of removing calibration data from other subpopulations, thus reducing data heterogeneity. As we observe from Table 12, as $\beta$ increases, the standard deviation of coverage increases which further supports the introduction of $\beta$ to reduce data heterogeneity. We also observe that as $\beta$ increases the mean of coverage approaches the ideal $1 - \alpha$ due to the fact that coverage guarantee of conformal prediction is a random quantity and the randomness comes from the sampling of the calibration set. Increasing $\beta$ increases the effective calibration set size which leads to more ideal coverage. We will refer to Section 3.2 of Angelopoulos & Bates (2021) for the full analysis of the effect of calibration set size. The exact value of $\beta$ to choose is task dependent and require some analysis of data in the embedding space for each task.

For $\sigma$, it is temperature scaling factor when converting embedding similarities to a probability distribution. From Table 12, we observe that lower $\sigma$ leads to lower standard deviation of coverage across the test environments but less ideal mean. Due to the fact that the cosine similarities between the test data and calibration data are mostly less than 0.5 (see Figure 7), smaller $\sigma$ leads to more weight on the point mass at $\infty$, leading to lower $\hat{q}_\alpha$ and thus higher mean coverage. On the other hand, as $\sigma$ increases, standard deviation increases while mean decreases since Algorithm 3 reduces to unweighted conformal prediction as $\sigma \to \infty$. The exact $\sigma$ to use is again task dependent and acts as a trade-off between mean and standard deviation.

## H COMPUTE RESOURCES

An A40 GPU and 60GB of memory were used to compute all results or train the models. For the domain classifier, With a batch size of 32, the training took 15 hours for the 26 domain case and 48 hours for the 15 domain case.

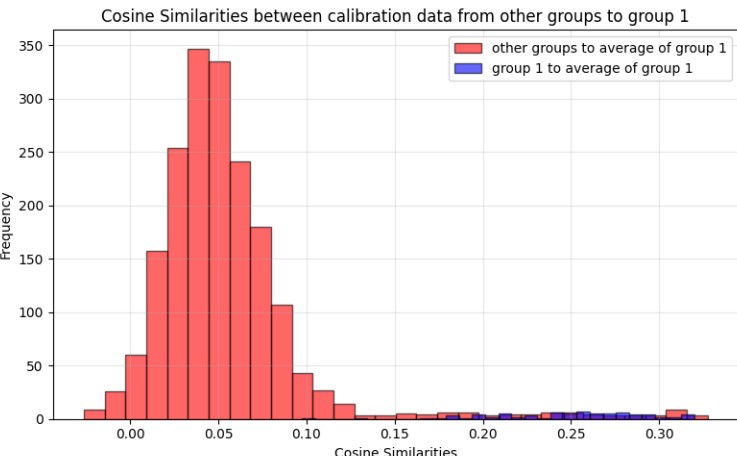

Figure 8: Average cosine similarities between embeddings of calibration data and test data. We observe that the calibration data embeddings from a domain has higher average similarities with test data embeddings from that same domain.

Table 12: Algorithm 3 coverage at $\alpha = 0.05$ with 26 domains and 3 classes per domain across different values of parameter $\sigma$ and $\beta$. Vision transformer is used as both the domain classifier and pretrained model with *LAC* as the score function. Mean and standard deviation of coverage across 100 test environments are shown.

| $\sigma$ | $\beta$ | | | | |
|---|---|---|---|---|---|
| | 0.05 | 0.15 | 0.25 | 0.35 | 0.45 |
| 0.6 | $0.982 \pm 0.005$ | $0.965 \pm 0.010$ | $0.961 \pm 0.012$ | $0.958 \pm 0.013$ | $0.957 \pm 0.013$ |
| 0.8 | $0.974 \pm 0.007$ | $0.962 \pm 0.011$ | $0.958 \pm 0.012$ | $0.957 \pm 0.013$ | $0.956 \pm 0.013$ |
| 1.0 | $0.971 \pm 0.007$ | $0.960 \pm 0.011$ | $0.957 \pm 0.013$ | $0.956 \pm 0.014$ | $0.955 \pm 0.014$ |
| 1.2 | $0.967 \pm 0.008$ | $0.959 \pm 0.011$ | $0.956 \pm 0.013$ | $0.956 \pm 0.014$ | $0.955 \pm 0.014$ |
| 1.4 | $0.965 \pm 0.009$ | $0.959 \pm 0.011$ | $0.956 \pm 0.013$ | $0.955 \pm 0.014$ | $0.955 \pm 0.014$ |
| 1.6 | $0.965 \pm 0.009$ | $0.958 \pm 0.011$ | $0.956 \pm 0.013$ | $0.955 \pm 0.014$ | $0.955 \pm 0.014$ |
| 1.8 | $0.965 \pm 0.009$ | $0.958 \pm 0.012$ | $0.956 \pm 0.013$ | $0.955 \pm 0.014$ | $0.954 \pm 0.014$ |
| 2.0 | $0.965 \pm 0.009$ | $0.957 \pm 0.012$ | $0.956 \pm 0.013$ | $0.955 \pm 0.014$ | $0.954 \pm 0.014$ |

