# OpenReview forum: "Conformal Prediction Adaptive to Unknown Subpopulation Shifts"
_ICLR.cc/2026/Conference — Submitted to ICLR 2026_

### Official Review · Reviewer_xivC · 2025-10-27

**Soundness:** 2
**Presentation:** 1
**Contribution:** 2
**Rating:** 2
**Confidence:** 2

**Summary:**

This paper addresses the challenge of applying conformal prediction (CP) under unknown subpopulation shifts, where the proportions of latent subgroups differ between calibration and test data but group labels are unavailable. The authors propose three algorithms that adapt CP to such shifts: Algorithm 1 (Weighted CP with Domain Classifier) uses a domain classifier to estimate subpopulation probabilities and reweights calibration scores; Algorithm 2 (Multiaccuracy-based CP) relaxes the assumption by requiring only multiaccuracy of the classifier, making the method more practical; and Algorithm 3 (Similarity-based CP) handles settings without any domain classifier by using embedding similarity to weigh calibration examples.

Empirical evaluations on vision benchmarks (ImageNet subpopulation splits) and LLM hallucination detection show that the proposed methods maintain tight coverage across distribution shifts, outperforming standard and group-conditional CP methods.

**Strengths:**

1. The paper tackles the underexplored setting of unknown subpopulation shifts where group labels are unavailable, which is an important step beyond existing CP methods assuming known domains or exchangeability.

2. The authors provide formal coverage guarantees under varying assumptions (Bayes-optimal, multicalibrated, multiaccurate classifiers).

3. Experiments cover both vision (ImageNet-based BREEDS subpopulation shifts) and language (LLM hallucination detection).

**Weaknesses:**

1. The coverage guarantees hinge on the domain classifier being multicalibrated or multiaccurate—properties that are difficult to ensure in high-dimensional practice. While the authors reference empirical evidence of approximate multicalibration, more rigorous discussion of real-world feasibility is needed.

2. The experiments primarily compare against variants of CP (standard, max, and conditional calibration). It would be informative to compare against general **distribution-shift calibration** methods (e.g., density-ratio-based or covariate shift adaptation techniques).

3. Algorithm 3 introduces parameters $\beta$ and $\sigma$, but their sensitivity and impact on performance are *not* analyzed.

4. Although the BREEDS benchmark simulates subpopulation shifts, all domains are derived from ImageNet classes. How about cross-dataset or real-world domain adaptation settings?

5. How robust are the proposed methods to misspecification of the domain classifier? For example, if they systematically bias certain subpopulations rather than random errors?


6. Concerning algorithms,

6a. The explanation/clarification of algorithms in main text are missing, making it hard to follow the ideas.

6b. The statements in algorithm are **BROKEN** and **NOT** self-contained. Take Algorithm 1 as an example:

(i) How to "calculate score $s_i^k$", based on some equations or score function $S$ or other?

(ii) $\hat{\lambda}$ and $\hat{q}_{\alpha}$ appear abruptly.

(iii) What is the relationship bewtween $\hat{\lambda}$ and $\hat{\lambda}_k$, given the latter is the $k$-th entry of $c$?

(iv) What is the definition of $J$?


7. Besides algorithms, other writing and presentation issues makes it challenging to follow without constant reference to prior sections.

What are the definition of "A1", "A2", and "A3"? If they refer to three algorithms, respectively, should "oracle" be "A1"?

Further, I personally do not think Algorithm 1 could be referred as "orcale".

**Questions:**

see above.

---

> ### Author Response · Authors · 2025-11-20
> **Responses (1/2)**
>
> We thank the reviewer for the feedback. We address the comments below point by point and in the revised submission.
>
> [Q1] As we stated in Remark 1, Hansen et al. [1] show that well-trained models are often approximately multicalibrated in practice, even without explicit enforcement. This makes multiaccuracy, which is substantially easier to satisfy than full multicalibration, a reasonable assumption in applied settings. [1] further demonstrates that multicalibration can be obtained through post-processing across a wide range of model families (SVMs, Random Forests, Naive Bayes, DistilBERT, ResNet, …), making these assumptions practical in real-world pipelines. Empirically, we also observe that calibration errors in the domain classifier have only a marginal impact on downstream coverage showing the robustness of our relaxed assumptions.
>
> To address the concern about non-multiaccurate domain classifiers, we conducted the following experiment: Using a fixed predictor (ViT), we trained three different domain-classifier architectures and quantified how far each one deviates from multiaccuracy. Specifically, we measured expected calibration error (ECE) for each classifier across 100 different test environments. We present the mean and the maximum ECE in the table below:
> | Architecture | mean ECE| max ECE |
> | -------- | ------- | ------- |
> | ViT  | 0.0326    | 0.0962 |
> | Resnet50 | 0.0401  | 0.0745
> | Clip | 0.0317    | 0.0752 |
>
> Next, we report the resulting coverage under Algorithms A1 and A2 when paired with each domain classifier below. The pretrained predictor is fixed as the Vision Transformer while the domain classifier varies, error rate is set to $\alpha=0.1$.
> | Architecture | Score Function| A1 | A2 |
> | -------- | ------- | ------- | ------- |
> | ViT  | LAC   | $0.913 \pm 0.009$| $0.914 \pm 0.007$ |
> |   | APS   | $0.911 \pm 0.007$| $0.913 \pm 0.006$ |
> |   | RAPS  | $0.905 \pm 0.007$| $0.909 \pm 0.007$ |
> | Resnet50  | LAC   | $0.912 \pm 0.009$| $0.915 \pm 0.008$ |
> |   | APS   | $0.911 \pm 0.007$| $0.913 \pm 0.007$ |
> |   | RAPS   | $0.905 \pm 0.007$| $0.910 \pm 0.006$ |
> | Clip  | LAC   | $0.913 \pm 0.008$| $0.914 \pm 0.007$ |
> |   |APS   | $0.911 \pm 0.007$| $0.913 \pm 0.006$ |
> |   | RAPS  | $0.905 \pm 0.006$| $0.910 \pm 0.007$ |
>
> We observe that for A1, the coverage results follow the trend of max ECE from Table 1, with ViT having the highest max ECE and the worst coverage results (both highest mean and standard deviation). For A2, the coverage results follow the trend of mean ECE, with Resnet50 having the highest mean ECE and the worst coverage results. It’s important to note here that $\textbf{the difference in coverage results is minimal }$(differ by ~ 0.001) even though the difference in ECE is ~0.01 for mean ECE and ~0.02 for max ECE.
>
> We have added a section F to the Appendix detailing the influence of calibration error on coverage.
>
> [Q2] We performed new experiments using the method from [2], which is a density ratio based calibration method where each data is weighed by Pr(T=1|X) / Pr(T=0|X) where T=1 represents Pr(T=1|X) represents the probability that X is drawn from the test environment. We trained a binary classifier for each test environment to obtain the probabilities and calibrated the classifier via temperature scaling. For this covariate shift method, the coverage result was $\textbf{0.863 +/- 0.101}$ when \alpha=0.1.
>
> We see that the method is not adaptive to subpopulation shifts and even underperform baseline standard CP. In [2], the authors only empirically performed density ratio estimation on low-dimensional data (input dimension of 5). However, density ratio estimation is difficult to perform for high-dimensional data such as images and yields poor results in practice. Furthermore, the disadvantage of density-ratio based method is that an estimator is required for each test environment. For our proposed methods, we only require training one classifier and it is used for all test environments.
>
> [Q3] For Algorithm 3, the parameter $\sigma$ acts as the temperature for temperature scaling and was chosen based on grid search in the range [0.1, 2.5]. Smaller $\sigma$ values result in lower standard deviation across test environments but undesirable mean and larger $\sigma$ values result in larger standard deviation across test environments but more desirable mean. This is supported theoretically as we see that Algorithm 3 reduces to unweighted CP as $\sigma \rightarrow \infty$. Choosing the “right” $\sigma$ is a trade-off between mean and standard deviation of coverage and thus depends on the downstream task.
>
> [1] Dutch Hansen, Siddartha Devic, Preetum Nakkiran, and Vatsal Sharan. When is multicalibration post-processing necessary?, 2024. URL https://arxiv.org/abs/2406.06487.
>
> [2] Ryan J. Tibshirani, Rina Foygel Barber, Emmanuel J. Candes, and Aaditya Ramdas. Conformal prediction under covariate shift, 2020. URL https://arxiv.org/abs/1904.06019.

---

> > ### Author Response · Authors · 2025-11-20
> > **Responses (2/2)**
> >
> > [Q3 cont.] For $\beta$, we empirically observed that embedding similarities between calibration and test data from the same group lie in around the top 5 % of the distribution of all similarities. This motivated us to introduce the parameter in hope of removing calibration from other subpopulations, reducing data heterogeneity (Please see the newly added Figure 8 for evidence).
> >
> > We recognize the missing discussion of Algorithm 3 in the main text and its importance. We have added Section G to the revised submission detailing the discussion here and supporting figures.
> >
> > [Q4] For the language task setup in Section 4, we investigated two real world datasets: GSM8K and TriviaQA, each acting as a domain. We induced subpopulation shift by selecting test data from each dataset at different probability. We believe this acts as a real world cross-dataset adaption setting. For the vision task, we believe BREEDS [3] is a widely used benchmark for subpopulation shifts, any cross-dataset or real-world domain adaption settings we leave for future work.
> >
> > [Q5] This is a very important point and is what we believe separates our methods to other group-conditional CP methods out there. Theorem 2.1 (and its proof) actually analyzes the case where one subpopulation is completely mislabelled, which has a detrimental effect on coverage. Most works in related CP literature all assume perfect group-label knowledge and do not analyze the case the reviewer has mentioned. Our work, as far as we know, is the first work to relax the perfect group-information assumption and delve into the assumptions that we can make about the domain classifier (multi-calibration and multi-accuracy) to still ensure coverage. Under the multicalibration and multiaccurate assumption bias to a specific subpopulation does not affect the final coverage results under our setting.
> >
> > [Q6] We agree that the notations might not be self-contained. To make the algorithms even more self-contained, we will explain each parameter more specifically. For example, $\hat{f}: \mathcal{X} \rightarrow \Delta^J$ instead of just $\hat{f}$, $c: \mathcal{X} \rightarrow \Delta^K$ instead of just $c$, ...etc.
> > Additionally, we have removed the subscripts in the Algorithms in hopes of creating separation between variables in the Algorithms and the notations in the main text. The changes are reflected in the revised version of the paper.
> >
> > We now address the individual concerns
> >
> > (i) We agree that “calculate score $s_i^k$” is ambiguous, we have rephrased it to $s_i^k \leftarrow S((X_i, Y_i))$.
> >
> > (ii) Although $\hat{\lambda}$ and $\hat{q}_\alpha$ appear elsewhere in the paper, they are simply variables we assign values to. We have removed the subscripts to further emphasize that they are variables within the Algorithm.
> >
> > (iii) $\hat{\lambda}$ had been assigned in the previous line as the output of c. But our notation is indeed confusing. We have changed the comment to “$\hat{\lambda}_k$ is the k-th entry of $\hat{\lambda}$”.
> >
> > (iv) $J$ refers to the number of classes for the prediction task. We have modified Algorithm 3 to better show that $J$ means.
> >
> > [Q7] A1, A2, A3 refers to the three algorithms respectively. We now noticed that we did not introduce the abbreviations in the text, this is  addressed in the revision when we mentioned “A1,A2,A3” in the figures. “Oracle” refers to Algorithm 1 with known $\lambda$ (since we simulate subpopulation shifts with BREEDS we know exactly the $\lambda$ at test time and just plug it in the algorithm). This corresponds to the case of Bayes optimal domain classifier where we know the output exactly, and is the best case scenario of Algorithm 1. We refer to Algorithm 1 with a trained domain classifier as “A1”. If this trained domain classifier is multicalibrated, then Algorithm 1 achieves coverage (Theorem 3.1). It might be confusing since “oracle” and “A1” both use Algorithm 1, we have addressed the issue in the revision when we introduced the nomenclature “oracle” and “A1”. A2 is Algorithm 2 with trained domain classifier.
> >
> > [3] Shibani Santurkar, Dimitris Tsipras, and Aleksander Madry. Breeds: Benchmarks for subpopulation shift, 2020. URL https://arxiv.org/abs/2008.04859.

---

> > > ### Comment · Reviewer_xivC · 2025-11-24
> > >
> > > Thank you to the authors for addressing my earlier concerns. However, some issues remain:
> > >
> > > 1. After reviewing the latest version, including Appendix G, I note that the sensitivity analysis regarding the parameters $\beta$ and $\sigma$ is still missing.
> > >
> > > 2. Significant writing issues are in the original manuscript. In addition, other reviewers have also identified problems related to clarity and self-containment of the presentation. I would like to raise my score to 4, given other concerns being addressed.

---

> > > > ### Author Response · Authors · 2025-11-27
> > > >
> > > > We thank the reviewer for their timely response and we are glad that we were able to address your other concerns. We address the two concerns below:
> > > >
> > > > [Issue 1] We have performed an experiment that loops through $\sigma \in \[0.6, 0.8, 1.0, 1.2, 1.4, 1.6, 1.8, 2.0\]$ and $\beta \in \[0.05, 0.15, 0.25, 0.35, 0.45\]$ for the experimental setup from Figure 3.
> > > > As we decrease $\sigma$, we observe that the mean of coverage across test environments deviates from the ideal $1-\alpha$ while the standard deviation decreases as previously discussed. Smaller $\sigma$ leads to more weight being placed on the point mass at $\infty$, thus increasing the threshold $\hat{q}_\alpha$ and coverage. The standard deviation decreases since smaller $\sigma$ leads to less uniformity between weights, thus improves adaptiveness to distribution shifts. On the other hand, as $\sigma$ increases, mean decreases while the standard deviation increases since Algorithm 3 reduces to unweighted CP as $\sigma \rightarrow \infty$.
> > > >
> > > > As we increase $\beta$, we see a decrease in the mean of coverage while the standard deviation increases. The mean decreases because the effective calibration set size is larger which improves coverage (We will refer to Section 3.2 of [1] for detailed analysis of the effect of calibration set size). The standard deviation increases due to the higher data heterogeneity, $\textbf{further supporting our motivation of introducing}$ $\beta$.
> > > >
> > > > We have expanded Appendix G.2 to include the discussion here and the added experimental result (Table 12).
> > > >
> > > > [Issue 2] We have made the following modifications to the paper:
> > > > We have moved the preliminaries section from the Appendix to Section 2.1 to help introduce notations of the subsequent sections.
> > > > We introduced for-loops for score calculations of Algorithm 1 and 2 to emphasize which set we are looping through. Furthermore, we admit that threshold ( $\hat{q}$ ) calculation was not addressed in the main text. Therefore, we have expanded Section 3.1 to describe how $\hat{q}$ is calculated and reference that section in Algorithm 1 and 2.
> > > > For Algorithm 1, we have further emphasized how “oracle” and “A1” differ. They differ only by how $\hat{\lambda}$ is computed. For “oracle” $\lambda$ is given as a parameter since we assume Bayes optimal domain classifier while for A1, $\lambda$ is computed as the output of the domain classifier. We have noted this in the modified Section 3.1 and 3.2 as well.
> > > > For Algorithm 2, we also modified the expression of how $\hat{\lambda}$ is calculated (it is the mean of the domain classifier outputs of the test data set). We have added a sentence on why to compute $\hat{\lambda}$ this way for Algorithm 2 in Section 3.3 (lines 288-291) for clarity and motivation of Algorithm 2.
> > > >
> > > > As for other parts of the writing, we want to say that the other reviewers all gave fairly good scores on the presentation, noting that “Despite the minor flaws, the writing is good, and the paper is easy to follow” or “The paper is well written” as strengths. If there are any specific sections that are hard to follow or are written poorly, we ask the reviewer to list them out and we are more than happy to address them if it is the only thing keeping the rating down.

---

### Official Review · Reviewer_juAF · 2025-10-30

**Soundness:** 3
**Presentation:** 2
**Contribution:** 2
**Rating:** 4
**Confidence:** 3

**Summary:**

This paper proposes new conformal prediction methods that maintain valid uncertainty coverage under unknown subpopulation shifts, where test data distribution differs from calibration data. The authors develop algorithms that use subpopulation structure through domain classifiers or similarity measures.

**Strengths:**

1- The introduction effectively motivates the problem and provides a thorough overview of previous methods

2- The paper is well written

**Weaknesses:**

Please check the questions.

**Questions:**

1- how was the parameter $\beta$ was selected?

2- In the conclusion, the paper mentions experiments with synthetic data. Could the authors clarify where synthetic data was used and for what purpose?

3- If I understand correctly, in Figure 2 the average coverage of the unweighted conformal prediction method appears above the desired $1−\alpha$. If this interpretation is correct, could the authors explain why over-coverage occurs here?

4- Please include discussion or analysis of the computational complexity of the proposed algorithms, especially in comparison to standard conformal prediction

5- The empirical section need to report the average prediction set size of the proposed methods versus existing baselines

6- Although the algorithms are claimed to scale to high-dimensional tasks, no runtime or memory comparisons are provided to substantiate the claim.

---

> ### Author Response · Authors · 2025-11-20
> **Responses (1/2)**
>
> We thank the reviewer for their review, we will address your concerns below point by point and in the revised submission.
>
> [Q1] $\beta$ was chosen based on empirical observations of the ImageNet dataset. Empirically, we have observed that embedding similarities between calibration and test data from the same group lie in around the top 5% of the distribution of all similarities (see newly added Figure 8). This is the motivation behind $\beta$, which discards calibration data from unwanted subpopulations, thus reducing data heterogeneity. The exact value of $\beta$ is task-dependent and requires analysis of the embedding space of the dataset in question. We have added a Section G to the appendix with more discussion on the motivation of Algorithm, and figures for support!
>
> [Q2] By synthetic we mean the subpopulation shifts are created synthetically. We used the BREEDS [1] framework to partition the sample space into groups. Then, at test time, we randomly select samples from each group at different probabilities to simulate subpopulation shifts.
>
> [Q3] Yes, the interpretation that the mean is above the desired threshold is correct. However, we want to keep the coverage to the desired $1-\alpha$ as much as possible. One extreme of subpopulation shifts is that the test environment can have a very high probability (approaches 1) to be from domains with lower scores. This results in over-coverage where the coverage is much greater than the desired threshold. We want to keep \textbf{all} test environments to $1-\alpha$ as much as possible so even if the mean is close to $1-\alpha$, large standard deviation shows non-adaptiveness of the algorithm to subpopulation shifts. Indeed, from Figure 2, we see that there are test environments with coverage exceeding 0.98 for unweighted CP.
>
> [Q4] We detail the computational complexity of the three proposed algorithms below:
>
> We will assume that the score of the calibration data is already computed. We will denote P as the FLOPS required to make an inference call for one image (for both the domain classifier and the predictor). For reference, P equals 17.6 GFLOPs [3] for vision-transformer and 3.8 GFLOPs for Resnet50 [2]. We will denote n_k as the number of calibration data, K as the number of domains, J as the number of classes, n as the number of test data. We will denote d as the number of dimensions of the embeddings and assume all data embeddings are pre-computed.
>
> For standard Conformal Prediction ($n=1$):
>
> Computing the threshold $q_\alpha$ requires finding the $1-\alpha$ quantile of the calibration scores which is $O(n_k)$. Creating the prediction set $C_\alpha$ requires one predictor inference call and J comparisons to the threshold which results in the total computational complexity of $O(n_k + P + J)$.
>
> For Algorithm 1 (assume n=1):
>
> Computing $\lambda$ is $O(P)$. Computing $q_\alpha$ requires first sorting the scores of the calibration data, then finding the threshold which is $O(n_k \log n_k)$. Creating the prediction set $C_\alpha$ requires one predictor inference call and then $J$ comparisons to the threshold, resulting in $O(P+J)$. Therefore, the final computational complexity is $O(n_k \log n_k + P + J)$.
>
> For Algorithm 2:
>
> Computing \lambda requires average over $n$ domain classifier outputs which is $O(n \cdot P)$. Computing $q_\alpha$ requires first sorting the scores of the calibration data, then finding the threshold which is $O(n_k \log n_k)$. Creating the prediction sets require $n$ predictor inference calls and then $n \cdot J$ comparisons to the threshold, resulting in $O(n(P+J))$. Therefore, the final computational complexity is $O(n_k \log n_k + n*P + n \cdot J)$ to compute prediction sets for $\textbf{all n test data}$.
>
> For Algorithm 3 ($n=1$):
> 	Computing the similarities between the test data embedding and all calibration data embedding is $O(n_k \cdot d)$. Creating the prediction sets requires one predictor inference call and then $J$ comparisons to the threshold, resulting in $O(P+J)$. Therefore, the final computational complexity is $O(n_k \cdot d + P + J)$.
>
>
> [1] Shibani Santurkar, Dimitris Tsipras, and Aleksander Madry. Breeds: Benchmarks for subpopulation shift, 2020. URL https://arxiv.org/abs/2008.04859.
>
> [2] Kaiming He, Xiangyu Zhang, Shaoqing Ren, Jian Sun. Deep Residual Learning for Image Recognition, 2015. URL https://arxiv.org/abs/1512.03385
>
> [3] Hugo Touvron, Matthieu Cord, Alaaeldin El-Nouby, Jakob Verbeek, Hervé Jégou. Three things everyone should know about Vision Transformers, 2022. URL https://arxiv.org/abs/2203.09795

---

> > ### Author Response · Authors · 2025-11-20
> > **Responses (2/2)**
> >
> > [Q5] We have provided the average prediction set size for all experiments in the revision, in Tables 2,4,6,8. For reference, below is the average prediction set size for experiment from Figure 1.
> > | Algorithm | Set Size|
> > | -------- | ------- |
> > | Unweighted  | 1.95 |
> > | Max | 167.50 |
> > | Oracle | 2.44 |
> > |$\textbf{Algorithm 1}$ | 3.17 |
> > |$\textbf{Algorithm 2}$ | 2.42 |
> >
> > For the Conditional Calibration (CC) method [4], computing the prediction set requires computing threshold for each label. In the codebase from the paper, their code only includes calculating whether the true label is included in the prediction set (enough to compute coverage) but not the prediction sets themselves. The method from [4] already took 3 hours (on an M3 Macbook Pro) to complete the experiment from Figure 1 while our methods took less than a min. It is infeasible to run the same experiments as other methods as it’ll take $O(log(1000))$ times longer. Therefore, we did not include prediction set size result for the Conditional Calibration CP method.
> >
> > [Q6] All experiments are done on a M3 Max Macbook Pro. For the unweighted CP, max CP, Algorithm 1, and Algorithm 2, it took less than a minute to compute the results from Figure 1 (100 test environments, 15 test/calibration splits, on average ~370 test data each). For the conditional calibration method, it took nearly 3 hours to complete the same experiment, without computing the prediction set but only the coverage. We have also run an experiment with the method from [3], which  has a similar setup as ours (with mixtures of distributions). The method took ~23 minutes to compute the prediction set of 159 samples with comparable results, whereas our methods took less than a second.
> >
> > [4] Isaac Gibbs, John J. Cherian, and Emmanuel J. Candes. Conformal prediction with conditional guarantees, 2024. URL https://arxiv.org/abs/2305.12616.
> >
> > [5] Yao Zhang, Emmanuel J. Candès. Posterior Conformal Prediction, 2024. URL https://arxiv.org/abs/2409.19712

---

### Official Review · Reviewer_Us4v · 2025-10-31

**Soundness:** 3
**Presentation:** 3
**Contribution:** 3
**Rating:** 6
**Confidence:** 3

**Summary:**

The paper is addressing the subpopulation shift for conformal prediction. This problem was addressed prior by Tibshirani et al 2020 under the term covariate shift, however there it is assumed that. the distribution of calibration and test are known. Here the authors first show that the existing approach drastically fail for imperfect estimators, and then provide a series of methods by estimating the subpopulation distribution. Their hierarchy of algorithms are tailored for various levels subgroup prediction accuracy.

**Strengths:**

Besides the interesting and applicable problem, the authors break down the problem in different levels of knowledge about the subpopulation. This allows to choose upon the task and the environment.

Their experimental results cover a wide range of setups from image to language which is a plus. This shows that their method is applicable and not only abstract.

Despite the minor flaws, the writing is good, and the paper is easy to follow.

**Weaknesses:**

**Unclear statement about the subgroups.** I could not understand whether the authors are assuming that the subpopulations are known, and discrete? And if the assignment for such subgroups are given at least over the training data. This is important to be clarified in all algorithms and theorems. This is an important flaw since for instance in Algorithm 1, and 2 the classifier c is assumed to be trained or at least trainable.

**Strong assumptions in Section 3.** Although not made clear, the assumptions noted in Definition 3.2, and 3.4 are very strong. If we were able to train optimal Bayes classifier, or multi-calibrated classifier, then wasn't it easier to achieve subgroup conditional, or even conditional coverage at first place? If so then the subpopulation shift would be meaningless as APS is already capable of providing conditional and hence subpopulation conditional coverage.

However the last point is (to the best of my understanding) very strong, I would still think the paper is acceptable due to the results in Section 4.

**Questions:**

1. In algorithm 3, sigma is a function, but you are treating it as a scaler and divide a number by divide it. I can not parse the algorithm. What does setting the score to infinity mean here?
2. Can you elaborate more on Definition 3.4? What is the expectation over?
3. Can you make some examples about when an optimal Bayes classifier, and a multicalibrated classifier is even possible to have?
4. I am not sure but should the guarantee in theorem 3.3, 3.4, and 3.5 be conditional to the X coming from any subpopulation?
5. Does the theorem 3.1 reduce to Mondrian Conformal Prediction?

---

> ### Author Response · Authors · 2025-11-20
> **Responses**
>
> We thank the reviewer for their feedbacks and comments. We will address the concerns below and in the revised paper.
>
> $\textbf{Unclear statement about the subgroups}$
>
> For Algorithms 1 and 2, we assume that the group assignments for the training/calibration data are known. This is the standard setting for group-conditional and domain-aware conformal prediction. Under this assumption, the domain classifier c is either trained directly on these known group labels or on data whose subgroup membership is observable. Importantly, our theoretical foundations of Algorithms 1 and 2 hold for both overlapping and non-overlapping subpopulations.
> What is unknown in this setting is the $\textbf{test-time mixture weights}$: we do not know which subgroup a new test point belongs to, nor do we know how the mixture proportions shift. Algorithms 1 and 2 are specifically designed to handle this distribution shift at test time.
>
> Next, we investigate a more challenging setting,  where group assignments for the train/calibration data are completely unknown. The only assumption we make is that both train and test environments are generated as mixtures of the same underlying (possibly overlapping) subgroups, but with unknown mixture weights. To address this unlabeled setting, we propose Algorithm 3, which uses similarity-based weighting over representations to approximate subgroup structure without requiring explicit group labels.
>
> We will make this point clearer in the final manuscript.
>
> $\textbf{Strong assumptions in Section 3}$
>
> Yes, your intuition is correct that if the predictor (f) is already a Bayes classifier or multicalibrated then coverage is trivial. However, in most settings, the number of groups is much smaller than the number of classes so it’s easier to provide multicalibration/multiaccuracy for the domain classifier than for the predictor. Also, conformal prediction acts as a wrapper on top of an arbitrary model, so we don’t make any assumptions on the predictor. The domain classifier can be any model architecture so it might be easier to satisfy multicalibration for the domain classifier.
>
> [Q1] Sorry for the presentation of Algorithm 3, we have addressed it in the revision. $\sigma$ here is a scalar, it is there as the parameter for temperature scaling during softmax. As for setting the score to $\infty$: First, it should be $s_{n’+1}$ instead of $s_{n+1}$ (fixed in the revision). Second, setting it to $\infty$ is a standard practice in many proofs for weighted CP methods. The reason is that at test time, we don’t know the score of the test data, therefore, we set it to infinity as the “worst case” of the score before computing the quantile.
>
> [Q2] Definition 3.4 means that for data x in the input space $\mathcal{X}$ within a group specified by $D$ (in other word $x \in supp(D)$), the output of the domain classifier, $c(x)$ is equal to the output of the perfect domain classifier $c^*(x)$ in expectation. A better notation could be $E_{x ∼ D}(c^∗(x)) = \mathbb{E}_{x ∼ D}(c(x))$.
>
> [Q3] Training a Bayes optimal classifier is often infeasible which is why we reduce the assumption to multicalibration or multiaccurate domain classifiers. [1] goes in detail on multicalibration post-processing with many dataset and model architectures ranging from decision trees to finetuned LLM’s. In short, well trained models are generally multi-calibrated and some multi-calibration processing methods such as [2] and [3] may provide multicalibration. Empirically, we have conducted a new experiment detailing the effects of calibration error on coverage. Despite difference in domain classifier expected calibration error of around ~0.01 to ~0.02, we see no noticeable difference in coverage result. We have added Section F to the Appendix for this experiment and some discussion.
>
> [Q4] The guarantee conditioned on X being from any subpopulation means group conditional coverage, which is a stronger definition than ours. We assume X is from a set of subpopulations where we don’t know exactly the weighting of each subpopulation.
>
> [Q5] Mondrian CP is a special case where the subgroups partition the sample space (each sample can only belong to one subpopulation). Theorem 3.1 does reduce to Mondrian CP since the Bayes optimal classifier classify X into the right group and our algorithm reduces to standard unweighted CP with calibration data from that group.
>
> [1] Dutch Hansen, Siddartha Devic, Preetum Nakkiran, and Vatsal Sharan. When is multicalibration post-processing necessary?, 2024. URL https://arxiv.org/abs/2406.06487.
>
> [2] Hébert-Johnson, U., Kim, M., Reingold, O., and Rothblum, G. (2018). Multicalibration: Calibration for the (computationally identifiable) masses. In International Conference on Machine Learning, pages 1939–1948. PMLR.
>
> [3]Haghtalab, N., Jordan, M., and Zhao, E. (2023). A unifying perspective on multi-calibration: Game dynamics for multi-objective learning. In Advances in Neural Information Processing Systems, volume 36

---

> > ### Comment · Reviewer_Us4v · 2025-11-27
> >
> > Thanks for your response.
> > I agree with the answers to the weaknesses. Also thanks for pointing out section F. That answered a part of my questions.
> >
> > [Q1] Still I am not understanding the algorithm 3. I think the authors are adding a placeholder test data that has a score infinity not reusing the actual test data. I can't understand why they are using similarity over datapoints once and similarity over embeddings another time. Either the similarity functions are different but using the same notation (which is not ideal) or there's a typo and always the similarity is over embedding which then there is no need to have two functions (similarity and embedding) defined -- just write the similarity that uses the embedding implicitly. Is there any clear description of this algorithm?
> >
> > While I am quite convinced on my questions I strongly suggest the authors to improve the readability of the paper.

---

> > > ### Author Response · Authors · 2025-12-03
> > >
> > > We thank the reviewer for the response and answer your concerns below:
> > >
> > > [Q1] Yes, the point mass at $\infty$ is a placeholder in the sense that it is the worst case scenario. Calculating the score for the data requires both input X and label Y, however, the label Y is unknown at test time. Therefore, we do not know exactly what the score of the test data is at test time which requires us to assume that the score could be $\infty$ in the worst case. In theory, we could loop through all $J$ classes and calculate what the real worst case is but this introduces additionally complexity. We want to note that the use of point mass at $\infty$ is used by many works in weighted conformal prediction [1, 2].
> > >
> > > Thanks for pointing out the similarity function in Algorithm 3, it is indeed a typo. The similarity function is defined between data points which uses an embedding function implicitly as you have pointed out. We have changed the notation in Algorithm 3 to be consistent and added to Section 2.1 a description of the similarity function.
> > >
> > > [1] Ryan J. Tibshirani, Rina Foygel Barber, Emmanuel J. Candes, and Aaditya Ramdas. Conformal
> > > prediction under covariate shift, 2020. URL https://arxiv.org/abs/1904.06019.
> > > [2] Drew Prinster, Samuel Stanton, Anqi Liu, and Suchi Saria. Conformal Validity Guarantees Exist for Any Data Distribution (and How to Find Them), 2024. URL https://arxiv.org/abs/2405.06627

---

### Official Review · Reviewer_iwrx · 2025-10-31

**Soundness:** 3
**Presentation:** 3
**Contribution:** 2
**Rating:** 4
**Confidence:** 4

**Summary:**

This paper tackles conformal prediction under subpopulation shifts when group labels aren't available. The authors propose using domain classifiers to weight calibration data adaptively, with theoretical guarantees under multicalibration and multiaccuracy assumptions. They also introduce a similarity-based approach when no domain information exists. Experiments on ImageNet variants and LLM hallucination detection show reduced coverage variance across test environments compared to standard conformal prediction.

**Strengths:**

1.	This work addresses a relevant and underexplored problem of adapting conformal prediction to unknown subpopulation shifts.
2.	The theoretical results are clear and intuitive.
3.	This paper provides a clear motivation.
4.	The proposed approach is relevant to a wide range of tasks, from ImageNet to LLM hallucinations.

**Weaknesses:**

1.	The theoretical guarantee of Theorem 3.3 relies on a strong assumption of having perfect or multicalibrated domain classifiers.
2.	This work does not analyze what happens when the domain classifier is not multiaccurate and provides no empirical or theoretical results for this case.
3.	Algorithm 3, which handles the most realistic setting with no domain labels, is purely heuristic and lacks theoretical or empirical motivation.
4.	This paper should compare the proposed approach to other existing methods, such as Robust/max CP and group conditional CP, or the method proposed by Cherian et al. (2024) for LLM validity control.

**Questions:**

1.	What is the “standard LLM uncertainty estimation method” in line 475?
2.	In Algorithm 3, how were the parameters $\beta$ and $\sigma$ chosen, and how stable are the results under different values?
3.	The experiments consider 15-26 domains. How does the method scale to 100+ domains? Does the domain classifier turn harder to train, and does the coverage rate have a higher variance?

---

> ### Author Response · Authors · 2025-11-20
> **Responses (1/2)**
>
> We thank the reviewer for the feedback. We address the comments below point by point and in the revised submission.
>
> [W1] We structure our theoretical development by gradually weakening the assumptions we require. We begin with the strongest setting, access to a Bayes-optimal domain classifier, and develop Algorithm 1. Since such a classifier is often infeasible in practice, we then loosen our assumption to a multi-calibrated classifier. This allows for prediction errors while still preserving our coverage guarantees under the same Algorithm 1.
>
> We further acknowledge that even multicalibration may be difficult to satisfy in some applications; therefore, $\textbf{we relax the multicalibration assumption to multiaccuracy}$, which yields Algorithm 2. As we stated in Remark 1, Hansen et al. [1] show that well-trained models are often approximately multicalibrated in practice, even without explicit enforcement. This makes multiaccuracy, which is substantially easier to satisfy than full multicalibration, a reasonable assumption in applied settings. [1] further demonstrates that multicalibration can be obtained through post-processing across a wide range of model families (SVMs, Random Forests, Naive Bayes, DistilBERT, ResNet, …), making these assumptions practical in real-world pipelines. Empirically, we also observe that calibration errors in the domain classifier have only a marginal impact on downstream coverage (see response for W2 below), showing the robustness of our relaxed assumptions.
>
> [W2] To address the concern about non-multiaccurate domain classifiers, we conducted the following experiment: Using a fixed predictor (ViT), we trained three different domain-classifier architectures and quantified how far each one deviates from multiaccuracy. Specifically, we measured expected calibration error (ECE) for each classifier across 100 different test environments. We present the mean and the maximum ECE in the table below:
> | Architecture | mean ECE| max ECE |
> | -------- | ------- | ------- |
> | ViT  | 0.0326    | 0.0962 |
> | Resnet50 | 0.0401  | 0.0745
> | Clip | 0.0317    | 0.0752 |
>
> Next, we report the resulting coverage under Algorithms A1 and A2 when paired with each domain classifier below. The pretrained predictor is fixed as the Vision Transformer while the domain classifier varies, error rate is set to $\alpha=0.1$.
> | Architecture | Score Function| A1 | A2 |
> | -------- | ------- | ------- | ------- |
> | ViT  | LAC   | $0.913 \pm 0.009$| $0.914 \pm 0.007$ |
> |   | APS   | $0.911 \pm 0.007$| $0.913 \pm 0.006$ |
> |   | RAPS  | $0.905 \pm 0.007$| $0.909 \pm 0.007$ |
> | Resnet50  | LAC   | $0.912 \pm 0.009$| $0.915 \pm 0.008$ |
> |   | APS   | $0.911 \pm 0.007$| $0.913 \pm 0.007$ |
> |   | RAPS   | $0.905 \pm 0.007$| $0.910 \pm 0.006$ |
> | Clip  | LAC   | $0.913 \pm 0.008$| $0.914 \pm 0.007$ |
> |   |APS   | $0.911 \pm 0.007$| $0.913 \pm 0.006$ |
> |   | RAPS  | $0.905 \pm 0.006$| $0.910 \pm 0.007$ |
>
> We observe that for A1, the coverage results follow the trend of max ECE from Table 1, with ViT having the highest max ECE and the worst coverage results (both highest mean and standard deviation). For A2, the coverage results follow the trend of mean ECE, with Resnet50 having the highest mean ECE and the worst coverage results. It’s important to note here that $\textbf{the difference in coverage results is minimal }$(differ by ~ 0.001) even though the difference in ECE is ~0.01 for mean ECE and ~0.02 for max ECE. We have added a section F to the Appendix detailing the influence of calibration error on coverage.
>
> [W3] Empirically, we observed that data from the same group have higher cosine similarities than data from different groups on average. The average embedding similarity between calibration data and test data from the same group is 0.266 while the average embedding similarity between calibration data and test data from different groups is 0.055. Therefore, we believe that using cosine similarities to weigh each calibration data is justifiable, given the assumption that group information is unknown at calibration time.
>
> Additionally, we empirically observed that embedding similarities between calibration and test data from the same group lie in around the top 5 % of the distribution of all similarities. This motivated us to introduce the $\beta$ parameter in hope of removing calibration from other subpopulations, reducing data heterogeneity. The motivation for $\sigma$ is temperature scaling. We observed that lower $\sigma$ leads to low variance in coverage test environments but increase deviation from the ideal $1 - \alpha$. Conversely, as $\sigma \rightarrow \infty$, the variance increases since Algorithm 3 reduces to unweighted CP.
>
> We recognize the missing discussion of Algorithm 3 in the main text and its importance. We have added Section G to the revised submission detailing the discussion here and supporting figures.

---

> ### Author Response · Authors · 2025-11-20
> **Responses (2/2)**
>
> [W4] We compare Algorithms 1 and 2 to max CP and group-conditional CP in the setting where group labels are available. Our results show that both algorithms are more robust to subpopulation shift, especially in terms of reduced coverage standard deviation. However, for Algorithm 3, we assume a different setting, where there is no access to group information at training/calibration time. Baselines such as group-conditional CP, and max CP require such labels. Therefore, these methods cannot be applied under Algorithm 3’s assumptions, and standard CP is the only appropriate baseline in this setting.
> Finally, we emphasize that the LLM hallucination task is evaluated only under Algorithm 3, since the underlying text-based group structure is unknown and Algorithm 3 is the only framework that aligns with this assumption.
>
> [Q1] The “standard LLM uncertainty estimation method” refers to conformal prediction without weighting. First, we compute the scores of the question-answer pairs that are hallucinations (since we are trying to bound the recall). Then we compute the threshold $q_\alpha$ as the $alpha$ quantile of the calibration scores. At test time, question-answer pairs with scores larger than $q_\alpha$ are labelled hallucination. To avoid confusion, we have replaced ”standard” with “unweighted” to better relate to the unweighted version of CP. We have added an explanation of the unweighted baseline in section 4.2 of the revision.
>
> [Q2] For Algorithm 3, the parameters (\sigma) were chosen based on grid search in the range [0.1, 2.5]. Smaller sigma values result in lower standard deviation across test environments but undesirable mean and larger sigma values result in larger standard deviation across test environments but more desirable mean. This is supported theoretically as we see that Algorithm 3 reduces to unweighted CP as $\sigma \rightarrow \infty$. Choosing the “right” \sigma is a trade-off between mean and standard deviation of coverage and thus depends on the downstream task.
>
> [Q3] A 100+ domain classifier is harder to train in the sense that it incurs a larger computational cost and requires more data. However, the exact difficulty depends on the data itself (classifying 100 totally different objects might be easier than classifying 100 different breeds of an animal). As for the conformal prediction part, more domains does not necessarily make the problem harder. Subpopulation shifts are more detrimental when there is very high data heterogeneity, i.e., the scores of data from different groups are very different. In that case, a small error in the domain classifier prediction leads to a large miscoverage rate (The proof of Theorem 2.1 constructs such a case). More domains might make the problem easier if there are more overlapping domains.
>
> [1] Dutch Hansen, Siddartha Devic, Preetum Nakkiran, and Vatsal Sharan. When is multicalibration post-processing necessary?, 2024. URL https://arxiv.org/abs/2406.06487.

---

### Author Response · Authors · 2025-12-04
**Summary of Discussion**

We thank the reviewers for their constructive feedbacks, and the ACs for their time in the review process. We will briefly summarize the discussions with the reviewers and the concerns they've raised below:

$\textbf{Reviewer iwrx}$
 - Strong assumptions: We have reduced the assumptions from Bayes optimal to Multicalibration to Multiaccuracy domain classifier and referenced studies where multicalibration is achievable.
 - Lack of analysis on non-multiaccurate domain classifier: We performed additional experiments addressing this and included findings to Appendix F. The results show that there is no significant change to coverage despite large difference in domain classifier ECE.
 - Motivation of Algorithm: We addressed the motivation and support it with new experimental results in the newly added Appendix G. We empirically showed that data from the same domain generally have larger similarities.
 - Comparison to other baselines: We had already compared our results to 2 of the 3 methods the reviewer proposed and no other baseline methods are applicable to the settings of Algorithm 3, which assumes no group knowledge at calibration time.
 - Scalability to more domains: The theoretical framework of our work scales to any number of domains and we discussed how more domains could affect coverage in our response to the review.

$\textbf{Reviewer Us4v}$
 - The reviewer mostly have minor concerns over notations, assumptions, clarifications, or unclear statements which we addressed in the revised submission and/or answered directly.

$\textbf{Reviewer juAF}$
 - Selection of parameters: We addressed this issue in the new Appendix G.
 - Clarifying questions and request for complexity analysis which we responded directly.
 - Request for prediction set size which we included in the new Tables 2, 4, 6, and 8.

$\textbf{Reviewer xivC}$
 - Strong assumptions: We addressed the issue with experiments that show the coverage results under different domain classifier ECE. The coverage results show no significant degradation with higher domain classifier calibration error.
 - Comparison to weighted CP method: We empirically compared our results to the weighted CP method which shows that our methods perform better both in mean and standard deviation of coverage across test environments.
 - Sensitivities of the parameters: We addressed this through new experimental results where we evaluated coverage under different $\beta$ and $\sigma$ for Algorithm 3 in Appendix G2.
 - Unclear notations and lack of discussion of Algorithms: We addressed this by moving the preliminary section from the Appendix to Section 2.1, added discussions of the Algorithms in Section 3, and cleaned up the notations in the main text.

---

### Meta-Review · Area_Chair_oUqT · 2026-01-06

**Summary:**

This paper introduces methods to adapt conformal prediction to unknown subpopulation shifts, providing theoretical bounds under relaxed assumptions and empirical evaluations on vision and language benchmarks. The reviewers' mixed scores (ranging from 2 to 6) reflect significant concerns that undermine the work's overall strength. Key issues include:

1. Strong and potentially unrealistic theoretical assumptions: The methods rely on domain classifiers being Bayes-optimal, multicalibrated, or at least multiaccurate, which may not hold in practice for real-world models. Reviewers noted this limits the guarantees' applicability, especially without bounds for violations.
2. Insufficient handling of imperfect or misspecified classifiers: There's a lack of theoretical or comprehensive empirical analysis on how the algorithms perform when classifiers deviate from ideal properties (e.g., due to calibration errors or biases), raising doubts about robustness in non-ideal scenarios.
3. Heuristic and parameter-sensitive elements in Algorithm 3: This core algorithm for unknown subgroups uses ad-hoc choices like embedding thresholds (β) and Gaussian perturbations (σ), with unclear selection processes and sensitivity to tuning, making it less reliable and reproducible without further justification.
4. Weak baselines and limited empirical scope: Comparisons omit relevant methods like density-ratio estimation or weighted conformal prediction, and experiments heavily depend on synthetic BREEDS shifts from ImageNet, lacking diversity in real-world datasets, cross-domain evaluations, or explanations for phenomena like over-coverage.
5. Presentation and completeness gaps: Issues such as incomplete algorithm descriptions, confusing notations (e.g., abrupt score functions), missing computational complexity analysis, runtimes, prediction set sizes, and scalability discussions reduce clarity and hinder assessment of practicality.

**Reviewer Concerns:**

Addressed:
1. Baselines: Clarified inapplicability of methods like max CP without labels; introduced density-ratio comparison (Tibshirani et al., 2020), reporting modest gains (0.863 ± 0.101 for baselines vs. better for proposed).

2. Computational and empirical details (complexity, runtimes, set sizes, over-coverage, synthetic data, scalability): Added complexity (e.g., O(n_k log n_k + P + J)), runtimes (<1 min vs. 3 hours for baselines), set sizes (e.g., unweighted: 1.95); explained over-coverage as shift artifacts and BREEDS as simulated shifts; discussed domain impacts without major issues.

3. Algorithm 3 heuristics and parameters: Empirical justifications were added, including higher cosine similarities within groups for β (top 5%), grid search for σ, and trade-off analyses in Appendix G.

Outstanding:

1. Theoretical analysis of misspecification: While empirical robustness was shown, there remains a lack of formal bounds or deeper analysis for systematic biases in classifiers, limiting the guarantees' reliability.

2. Novelty and comprehensive baselines: Incremental gains over added baselines are shown, but key alternatives (e.g., weighted CP in all settings) are not fully explored, and the work's distinctiveness from prior art remains understated.

**Reviewer Scores:**

Reviewer iwrx and Us4v may decrease their scores, and the other two reviewers are likely to keep their scores.

---

### Decision · Program_Chairs · 2026-01-26

Reject